# Maternal obesity induces activator protein 1-mediated inflammatory response to impair embryonic neurogenesis

Li-Wei Chen[1] , Md Nazmul Hossain[1] , Yao Gao[1], Zhongyun Kou[1], Sharmeen Islam[1], Xinrui Li[1] , Chaeyoung Shin[1] , Jeanene Marie de Avila[1], Mei-Jun Zhu[2] and Min Du[1]

[1]*Nutrigenomics and Growth Biology Laboratory, Department of Animal Sciences, Washington State University, Pullman, Washington, USA*
[2]*School of Food Science, Washington State University, Pullman, Washington, USA*

Handling Editors: Laura Bennet & Justin Dean

The peer review history is available in the Supporting Information section of this article (https://doi.org/10.1113/JP289326#support-information-section).

**Abstract figure legend** Maternal obesity (MO) creates a pro-inflammatory embryonic environment, which activates the activator protein 1 (AP-1) transcription factor complex, resulting in reduced expression of key neurogenic regulators and impaired neuronal differentiation. Consequently neuronal populations decrease due to MO during early neurogenesis. Together these findings reveal an inflammation-driven transcriptional mechanism linking MO to disrupted embryonic neurogenesis. Figure created using BioRender.

**Abstract** Maternal obesity (MO) is a growing global problem, which poses significant risks to fetal neurodevelopment and long-term neurological functions of offspring, but the underlying molecular

**Li-Wei Chen** received an MS degree in animal science from National Chung Hsing University and is currently a PhD student in the Department of Animal Sciences at Washington State University. His research focuses on how maternal obesity influences embryonic brain development, with particular emphasis on inflammatory signalling and the role of the placenta in shaping offspring neurogenesis. Through this work, he has gained experience in computational analyses. He aims to extend this research across later developmental stages and into human datasets, with the long-term goal of translating findings from animal models into clinical strategies to improve child health outcomes.

mechanisms remain to be established. To address this female mice were fed either a control diet or a high-fat diet (HFD) for 2 months to induce obesity, and the same dietary treatments were maintained during pregnancy. Embryos were sampled at E11.5 and E13.5. Single-cell RNA sequencing revealed reduced proportions of neurons and neural progenitors in embryos from obese mothers. The down-regulation of neurogenesis, nervous system development and synaptic organization pathways were further confirmed by Gene Ontology analysis. Key neurogenic transcription factors, including *Neurod1*, *Neurog2* and *Ascl1*, were suppressed in MO embryos, accompanied by increased expression of inflammatory markers, including tumour necrosis factor-$\alpha$ (TNF-$\alpha$) and *Cxcl2*, and inflammatory signalling mediators, *Fos*, *Jun* and *Jund*. Single-cell ATAC sequencing revealed the activator protein 1 (AP-1) binding sites in the promoter regions of *Tnf* and *Cd68*, with MO-enhancing AP-1 transcription factor motif activity and increased chromatin accessibility in the loci of *Tnfa* and *Cd68* genes. Furthermore, TNF-$\alpha$ treatment of neurogenic cells suppressed *Neurod1* and *Neurog2* expression, suggesting a direct link between inflammatory signalling and impaired neurogenesis. Our findings suggest that MO creates a pro-inflammatory environment that disrupts neurogenesis during early embryonic development, providing new insights into the neurodevelopmental disorders in offspring born to obese mothers.

(Received 23 May 2025; accepted after revision 24 February 2026; first published online 14 March 2026)

**Corresponding author** Min Du: Nutrigenomics and Growth Biology Laboratory, Department of Animal Sciences, Washington State University, Pullman, WA 99164, USA. Email: min.du@wsu.edu

**Key points**

- Maternal obesity (MO) suppresses neurogenesis in the early embryos.
- Single-cell RNA sequencing (scRNA-seq) reveals decreased neurogenic cells and neurogenic factors in MO embryos.
- MO elevates activator protein 1 (AP-1) transcription factor accessibility to the promoters of inflammatory genes.
- Inflammation induced by tumour necrosis factor-$\alpha$ (TNF-$\alpha$) suppresses *Neurod1* and *Neurog2* expression and neurogenesis *in vitro*.

## Introduction

The obesity rate is increasing worldwide, including women at reproductive age. By the end of 2030, the prevalence of maternal obesity (MO) in developed countries is estimated to be 45%–50% (Muhammad et al., 2023). MO leads to several detrimental effects on fetal development, such as increased risks of metabolic disorders, impaired skeletal muscle development and neurodevelopmental disorders (Dearden et al., 2020; Godfrey et al., 2017). A higher risk of neuropsychiatric disorders, including attention deficit hyperactivity disorder (ADHD), autism spectrum disorders (ASD), depression and eating disorders, has been observed in offspring affected by MO (Batorsky et al., 2024; Edlow, 2017). MO induces a pro-inflammatory condition inside the uterus, which harms fetal brain development, though the underlying mechanisms remain poorly understood (Shook et al., 2020).

Neurogenesis mainly occurs in the prenatal and early postnatal stages. Neurodevelopment commences with the formation of the neural tube from the ectoderm, resulting in a pseudostratified epithelium consisting of neuroepithelial cells (NECs). During early embryonic development, NECs undergo symmetric divisions to expand their population, which then switch to asymmetric, producing NECs and radial glial cells (RGCs) by embryonic day 10 (E10) in the mouse telencephalon (Paridaen & Huttner, 2014). Obesity causes low-grade chronic inflammation, and maternal inflammation affects fetal neurogenesis by releasing pro-inflammatory cytokines that alter fetal brain environment (Han et al., 2022). The nuclear factor kappa-light-chain-enhancer of activated B cell (NF-$\kappa$B) signalling pathway is the main pathway activated under inflammation, which can further induce the expression of pro-inflammatory cytokines such as tumour necrosis factor-$\alpha$ (TNF-$\alpha$), interleukin

(IL)-1$\beta$ and IL-6 (Baker et al., 2011; Mohseni et al., 2023). These pro-inflammatory cytokines alter the expression of neurogenic genes and fetal brain development (Hsiao & Patterson, 2011). Inflammatory conditions activate early response genes, including FOS, JUN and JUND, which form the activator protein 1 (AP-1) transcription factor complex to mediate inflammatory responses (Hannemann et al., 2017). The AP-1 complex has been implicated in both inflammatory signalling and neural development (Kusnadi et al., 2019; Raivich et al., 2004).

The proliferation, differentiation and maturation of neural progenitor cells (NPCs) into neurons and glial cells are regulated by sequential expression of specific transcription factors. NEUROD1, NEUROG2 and ASCL1 are the basic helix–loop–helix family transcription factors that play distinct roles in the neurogenic process and central nerve system development. In the early stages of neurogenesis, the expression of *Pax6* and *Sox2* is essential for preserving the multipotency of NPCs (Ahmed et al., 2009; Sansom et al., 2009). In the initial phases of neurogenesis, NEUROG2 and ASCL1 initiate the expression of genes required for neuronal commitment and differentiation by stimulating the transcription of genes specific to neurons, including neurotransmitter biosynthesis enzymes, ion channels, synaptic proteins and proteins required for neuronal migration. In addition to activating neurogenic pathways, NEUROG2 and ASCL1 facilitate progenitor cells developing into neurons by suppressing gliogenesis, revealing their importance in the complex transcriptional networks required for the precise timing and co-ordination of neurogenesis (Hulme et al., 2022). Several brain regions, including the cerebral cortex, hippocampus and cerebellum, rely on NEUROD1 for proper development. It stimulates the production of genes required for the maturation and survival of neurons such as dendritic growth and synaptic function (Tutukova et al., 2021; Zhang et al., 2024). The correlation between NEUROD1 dysregulation and a range of neurological illnesses highlights its critical roles in preserving the intricate equilibrium between neuronal development and function (Chae et al., 2004).

Neurogenesis in mouse embryos mainly occurs between E11.5 and E17.5. Neurons generate and form distinct layers of the cortex during this period (Molyneaux et al., 2007). To examine the impacts of MO on the early neurogenesis, we performed single-cell RNA sequencing (scRNA-seq) of E11.5 and E13.5 embryos from control (CON) and obese mothers, and single-cell ATAC sequencing (scATAC-seq) was further used for analysing the genomic accessibility. We also evaluate the impacts of inflammation on neurogenesis *in vitro*. Finally we performed RT-qPCR analyses to analyse the neurogenic markers in E13.5 embryos. Data showed that MO suppressed neurogenesis, which is likely mediated by the inflammatory response in MO embryos.

## Materials and methods

**Animal treatments.** Eight-week-old female C57BL/6J mice (000664, The Jackson Laboratory, Bar Harbor, ME, USA) were randomly assigned to MO and CON groups. The CON diet (10% energy from fat, D12450H, Research Diets, New Brunswick, NJ, USA) and high-fat diet (45% energy from fat, D12451, Research Diets) were fed *ad libitum* to mice, respectively, to form the CON and MO groups. Mice had free access to water throughout the experiment. Obesity was confirmed when the MO group exhibited an average body weight at least 20% greater than that of CON mice (Gao et al., 2024). After obesity induction, C57BL/6J male mice maintained on regular chow were used for mating with experimental females. The presence of a vaginal plug in the early morning was designated as E0.5. Pregnant mice were anaesthetized by carbon dioxide inhalation and killed by cervical dislocation at E11.5 and E13.5. During pregnancy maternal mice continued their respective diets until embryo collection at E11.5 and 13.5, which were processed for single-cell suspension and scRNA-seq. At E13.5, brains were dissected for RT-qPCR and neural stem cell isolation. All animal studies were conducted in Association for Assessment and Accreditation of Laboratory Animal Care (AAALAC)-approved facilities and approved by the Institutional Animal Care and Use Committee at Washington State University (permit no. 6704).

**Maternal metabolic phenotype.** Maternal blood was collected at E13.5 by cardiopuncture under carbon dioxide anaesthesia. Blood glucose levels were measured by a glucose monitor. Serum was collected, and insulin level was measured using a Mouse Insulin ELISA Kit according to the manufacturer's instructions (cat. no. 10-1247-10, Mercodia, Uppsala, Sweden). Serum lipid profiles, including high-density lipoprotein cholesterol (HDL-C), low-density lipoprotein cholesterol (LDL-C) and triglycerides (TG), were analysed using commercial assay kits following the manufacturer's instructions (cat. no. NBP3-25823, NBP3-25881 and NBP3-24542, Novus Biologicals, Centennial, CO, USA). TNF-$\alpha$ levels were measured using the Mouse TNF alpha Uncoated ELISA Kit according to the manufacturer's instructions (cat. no. 88-7324-22, Invitrogen, Carlsbad, CA, USA).

**Embryo collection and scRNA-seq.** We confirmed the Theiler stage by counting somite numbers under a dissecting microscope (Leica DFC450C) to minimize developmental variations. For E11.5 and 13.5 embryos, Theiler stage 19 (45–47 somites) and stage 21 (52–55 somites) were selected, respectively, and the scRNA-seq was performed on whole embryos as previously described (Hossain et al., 2024). Mouse embryo contains key neurogenic structures, including the spinal neural tube

and neural crest-derived populations. The spinal neural tube at E11.5–E13.5 remains highly proliferative and actively produces neurons and neural progenitors, and therefore constitutes a relevant neurogenic compartment for assessing early neurodevelopmental processes (Saade & Marti, 2025). For each time point, 15 embryos from three independent litters of MO or CON mice were pooled together for cell suspension preparation. Embryo samples were minced and incubated with TryLE Express dissociation reagent (Life Technology) at 37°C for 15 min, with gentle agitation every 2 min. The reaction was quenched with heat-inactivated serum. The dissociated cells were washed, resuspended in PBS containing 0.04% bovine serum albumin (BSA) and filtered through a 40-μm cell strainer. After viability assessment and cell counting, single-cell suspensions were processed using the chromium controller. cDNA libraries were constructed using the Chromium Single Cell 3' Library & Gel Bead Kit, version 3, following the manufacturer's protocol (10× Genomics, Inc.). Samples were multiplexed and sequenced on an Illumina NovaSeq 6000 S4 platform (Hossain et al., 2025).

**scRNA-seq data processing.** Raw sequencing data were preprocessed using Cell Ranger, version 7.0 (10× Genomics), and aligned to the mouse mm10 transcriptome reference genome. Data integration was performed using the merge function in Seurat (version 5.2.0) to combine libraries from two different time points. Quality control was applied to remove low-quality cells and potential doublets by filtering cells with unique molecular identifier (UMI) counts exceeding 200,000 or expressing fewer than 500 genes. Additionally cells with mitochondrial gene content greater than 20% were removed. After quality control filtering the retained cells were processed for downstream analysis. The filtered dataset underwent normalization using the LogNormalize method with default parameters. Data scaling was performed, followed by linear dimensionality reduction using principal component analysis (PCA). Non-linear dimensionality reduction was performed using uniform manifold approximation and projection (UMAP) for visualization. Cell clusters were identified based on marker genes determined using the 'FindAllMarkers' function in Seurat. Differential gene expression analysis between two groups was conducted using the 'Wilcoxon rank-sum test' (Hossain et al., 2024; Zhao et al., 2021). Pseudotime trajectory analysis was performed using the principal curve method implemented in the princurve R package (version 2.1.6). UMAP dimensional reduction co-ordinates were extracted from the Seurat object containing and integrated with pseudotime values as a cell-level metadata feature. The resulting trajectory was visualized using FeaturePlot with cells coloured by their pseudotime values on a colour scale, and the fitted principal curve was overlaid to show the inferred path of cellular progression. To compare neural subpopulation cell numbers between the CON and MO groups, we normalized the MO cell counts to the CON group by dividing by 1.023, the ratio of MO to CON cell numbers (25 ,446 *vs.* 24 ,864 cells).

**Gene Ontology enrichment analysis.** Gene Ontology (GO) enrichment analysis was performed using the 'clusterProfiler' R package (version 4.12.6) and the org.Mm.eg.db mouse genome database. Differentially expressed genes (DEGs) with negative log2 fold change values were classified as downregulated and converted from gene symbols to ENTREZ IDs using the mapIds function. GO biological process enrichment analysis was conducted using the enrichGO function. The analysis parameters included Benjamini–Hochberg adjustment for multiple testing, with significance thresholds set at $P$-value <0.05. The top 10 significantly enriched GO terms were used for further analysis.

**Single-cell ATAC sequencing and data processing.** E13.5 embryos were collected and prepared for single-cell suspension at 100 cells/μL concentration. A total of $2 \times 10^6$ cells were collected by centrifugation of single-cell suspension at 300 $g$ for 5 min at 4°C. The pellet was resuspended in 200 μL lysis buffer (10 mM Tris-HCl, pH 7.4; 10 mM NaCl; 3 mM $MgCl_2$; and 1× PBS) and incubated on ice for 1 min. Nuclei were washed with Nuclei Wash Buffer (10× Genomics PN-CG000124), resuspended in Resuspension Buffer (10× Genomics PN-CG000124) and centrifuged at 500 $g$ for 10 min at 4°C twice (10× Genomics protocol CG000124). After the concentration assessment of the isolated nuclei, library preparation was performed according to the 10× Genomics single-cell ATAC-seq protocol.

Raw sequencing data were processed using Cell Ranger ATAC (version 2.1.0) and aligned to the mouse reference genome (mm10). Downstream analyses were performed using the Signac package (version 1.14.0) in R. Peak matrices for CON and MO groups were generated using the CreateChromatinAssay function with minimum thresholds of 10 cells and 200 features. Quality control metrics, including nucleosome signal, transcription start site (TSS) enrichment and percentage of reads in peaks, were calculated. The datasets were merged and normalized; dimensionality reduction was performed using singular value decomposition (SVD); and non-linear dimension reduction for visualization was conducted using UMAP. Clustering was performed using the FindNeighbors and FindClusters functions. To identify the cell clusters in scATAC-seq, transfer anchors were identified between the RNA expression matrix

and the ATAC-seq data using the FindTransferAnchors function in Seurat with canonical correlation analysis (CCA), and cell-type labels were transferred from the RNA-seq clusters to the ATAC-seq dataset. Integration results were visualized using dimensional reduction plots to compare the original RNA-seq clustering with the predicted cell types in the ATAC-seq data. Differentially accessible peaks between cell types were identified using the FindMarkers function with a log-fold change threshold of 0.2. AP-1 binding sites on *Tnf* and *Cd68* promoter region were identified using position frequency matrices (PFMs) from JASPAR with TFBSTools package. PFMs were converted to position weight matrices and used to scan the genomic sequence retrieved from BSgenome.Mmusculus.UCSC.mm10. Motif matching was performed using the searchSeq function, and putative binding sites were identified based on their relative scores >0.9.

**Motif activity analysis.** Motif activity analysis was performed on cluster 12, which was identified as an immune cell population based on scRNA-seq integration. Cells belonging to cluster 12 were sorted from the integrated ATAC-seq dataset. Transcription factor binding motifs were obtained from the JASPAR2020 CORE vertebrate collection using the TFBSTools package. Motif positions were identified using the motifmatchr package and BSgenome.Mmusculus.UCSC.mm10. Differential motif activity between CON and MO groups was assessed using the FindMarkers function in Signac, with chromVAR-derived accessibility deviations as the input matrix. Average differences in motif activity were calculated using rowMeans, and log2 fold changes were computed for each motif. Significantly differential motifs were identified based on their adjusted *P*-values and fold changes.

**Immunostaining.** E13.5 embryonic brains were fixed in 4% paraformaldehyde (PFA) at 4°C for 2 h and cryoprotected in sucrose at 4°C through a series of processes, including 30% sucrose overnight, a 1:1 mixture of 30% sucrose and optimal cutting temperature (OCT) compound overnight and OCT alone for 1 h. Samples were then embedded in OCT, frozen and cryosectioned at a thickness of 14 μm.

For antigen retrieval sections were immersed in boiling citrate buffer (10 mM citric acid, 0.05% Tween-20, pH 6.0) and steamed for 20 min. Sections were allowed to cool to room temperature and then blocked in blocking buffer (1× PBS containing 5% normal serum and 0.3% Triton X-100) for 60 min. Sections were incubated overnight at 4°C with a primary antibody against NeuN (24307, Cell Signaling Technology, Danvers, MA, USA). After three washes in 1× PBS (5 min each), sections were incubated

with the fluorochrome-conjugated secondary antibody for 1–2 h at room temperature in the dark. Sections were then washed thrice in 1× PBS (5 min each) and mounted with DAPI-containing mounting medium. For cell number quantification at least three sections from each individual were stained, imaged and analysed. All quantifications were performed using ImageJ with the 'Analyze Particles' function following uniform adjustment of brightness and contrast, with identical image processing parameters applied across all samples (Cui et al., 2020).

**Western blotting.** Proteins were extracted from collected frozen E13.5 brain tissues stored at –80°C using lysis buffer (100 mM Tris-HCl, pH 6.8, 2.0% SDS, 20% glycerol, 0.02% bromophenol blue, 5% 2-mercaptoethanol, 100 mM NaF and 1 mM Na$_3$VO$_4$) and separated as previously described (Hossain et al., 2025). The primary antibodies against TNF-$\alpha$ (3707) and JUN (9165) were purchased from Cell Signaling Technology (Danvers, MA, USA). $\beta$-Tubulin was used as a loading control and obtained from the Developmental Studies Hybridoma Bank (Iowa City, IA, USA). IRDye 800CW goat anti-rabbit and IRDye 680 goat anti-mouse secondary antibodies were purchased from LI-COR Biosciences (Lincoln, NE, USA).

**Chromatin immunoprecipitation-qPCR.** Chromatin immunoprecipitation (ChIP) was performed using the SimpleChIP Plus Enzymatic Chromatin IP Kit (Magnetic Beads, CST 9003, Cell Signaling Technology, Danvers, MA, USA) following the manufacturer's protocol. Briefly 25 mg of mouse embryo per immunoprecipitation (IP) was minced on ice and cross-linked in 1.5% formaldehyde. Tissue was disaggregated on ice with a Dounce homogenizer, nuclei was digested with micrococcal nuclease and lysed by sonication with 15 sets for 20 s on ice; 2% of input samples were collected until further analysis. For each IP 10 μg of digested, cross-linked chromatin was incubated overnight at 4°C with c-Jun Rabbit monoclonal antibody (CST 9165), Normal Rabbit IgG (CST 2729) or Histone H3 Rabbit antibody (CST 4620). Purified DNA was then analysed using real-time PCR with cycling, 95°C 3 min, 40 cycles of 95°C 15 sec and 60°C 60 s. The target locus is the JUN binding site on *Tnf* promoter with primers: forward 5'-ACCGCAGTCAAGATATGGCA-3' and reverse 5'-TGGGAATTCACGGACCTCAC-3'.

**Cell culture and cytokine treatment.** C57BL/6J mice were killed on E13.5, and embryos were removed and placed in sterile PBS. Brains were dissected and transferred to DMEM/F12 and triturated gently. The tissue was digested using 0.05% trypsin–EDTA for 5 min at 37°C, and 1 mL DMEM/F12 was added to stop digestion; the mixture was centrifuged at 1000 *g* for 5 min. After the supernatant was discarded the cell pellet was resuspended in

**Table 1. Primers used for RT-qPCR.**

| Gene name | Forward sequence | Reverse sequence |
| --- | --- | --- |
| *Neurod1* | 5′-CAGCTCAACCCTCGGACTTT-3′ | 5′-CTGGTGCAGTCAGTTAGGGG-3′ |
| *Neurog2* | 5′-CGTGGGGAACCTCGTAAGAC-3′ | 5′-GATTCACACGAACTGCCTGC-3′ |
| *Ascl1* | 5′-GTCCCCCTTTGATCGTGCTT -3′ | 5′-GGCTCCACTCTCCATCTTGC-3′ |
| *18s* | 5′-GTAACCCGTTGAACCCCATT-3′ | 5′-CCATCCAATCGGTAGTAGCG-3′ |

neural stem cell culture medium consisting of DMEM/F12 (1:1) supplemented with $1 \times B27$, 20 ng/mL of epidermal growth factor (EGF) and 20 ng/mL of basic fibroblast growth factor (bFGF). The cells were maintained at 37°C in a humidified incubator with 5% $CO_2$. For neuronal differentiation NSCs were seeded at a density of $5 \times 10^4$ cells per well in 12-well plates pre-coated with polyornithine and laminin (Sigma-Aldrich). Cells were cultured in neural stem cell culture medium for 2 days before switching to differentiation medium containing DMEM/F12 (1:1) supplemented with $1 \times B27$ (Yao et al., 2015). For cytokine stimulation $5 \times 10^4$ cells were seeded per well in polyornithine and laminin-coated 12-well plates and cultured for 2 days prior to TNF-$\alpha$ treatment. Cells were then treated with TNF-$\alpha$ at concentrations of 0, 10, 30 and 100 ng/mL in differentiation medium for 48 h. After treatment cells were washed twice with PBS, lysed with TRIzol and collected for subsequent qPCR analysis (Hagman et al., 2019).

**RT-qPCR.** Total RNA was isolated using TRIzol reagent (cat. no. 15596018, Invitrogen, Carlsbad, CA, USA) following the manufacturer's instructions. The iScript cDNA Synthesis Kit (Bio-Rad, cat. no. 1708891) was used to synthesize cDNA from 500 ng of purified RNA. Real-time quantitative PCR was performed using SsoAdvanced Universal SYBR Green Supermix (cat. no. 172570, Bio-Rad, Hercules, CA, USA) on a CFX RT-PCR detection system (Bio-Rad, Hercules, CA, USA). The relative mRNA expression levels were normalized to 18S ribosomal RNA and analysed using the 2-$\Delta\Delta$Ct method. All primer sequences used in this study are provided in Table 1.

**Statistical analysis.** For RT-qPCR embryos from six pregnant mice in each group were used ($n = 6$), and statistical significance was analysed using unpaired, two-tailed Student's $t$ test. For cell culture study three biological replicates were conducted, and data were analysed using one-way ANOVA for multiple comparisons. Data were analysed using GraphPad Prism, version 8.0.2 (GraphPad Software, San Diego, CA, USA). For scRNA-seq analysis 15 embryos from three litters were used at each time point. Differential expression analysis between treatment groups

was conducted using Student's $t$ test. All data are presented as mean $\pm$ SD, and statistical significance is $P < 0.05$.

## Results

**MO alters neural cluster composition of embryos.** The single-cell transcriptomic profiles were analysed by integrating all cells from whole embryos collected from two different time points, E11.5 and E13.5 (Fig. 1*A*). Maternal serum lipid profiles were assessed, including LDL, HDL and TGs, as well as measurements of glucose, insulin and the inflammatory marker TNF-$\alpha$. We found that obese dams exhibited significantly elevated levels of LDL, HDL, TGs and circulating TNF-$\alpha$ compared to control dams, indicating dyslipidaemia and systemic inflammation under MO conditions (Fig. A*2*). In contrast maternal glucose and insulin levels did not differ significantly between groups (Fig. A*2*). After low-quality cells were filtered out, 50,310 single-cell transcriptomes were retained, including 24,864 cells from the CON group and 25,446 cells from the MO group. The dataset comprised 25,521 genes, with a median of 5232 genes and 22,290 UMIs per cell. Dimensionality reduction was performed using PCA and UMAP for visualization, which identified 21 distinct cell clusters (Fig. 1*B*). Canonical marker analysis confirmed the identity of clusters (Fig. 1*C*; Fig. A1). Violin and feature plots illustrate the expression of neural markers, highlighting cluster-specific gene expression patterns. Neurons (cluster 5) prominently expressed *Dcx*, *Tubb3*, *Snap25* and *Rbfox3*, whereas neural progenitors (cluster 7) prominently expressed *Pax6*, *Fabp7*, *Rfx4* and *Sox2* (Fig. 1*D*; Figure S1). Particularly the proportion of neurons was reduced in the MO group (4.6%) compared to the CON group (9.7%), and neural progenitors were also reduced in the MO group (2.1%) compared to the CON group (7.3%), suggesting altered neurogenic cell populations under MO (Fig. 1*E*).

**Neural populations and developmental trajectories.** The neural cell populations were further analysed to understand the trajectory. Through re-clustering of neural populations eight subclusters were identified (Fig. 2*A*). The developmental trajectory analysis (Fig. 2*B*–*D*) illustrated the progression from progenitor cells

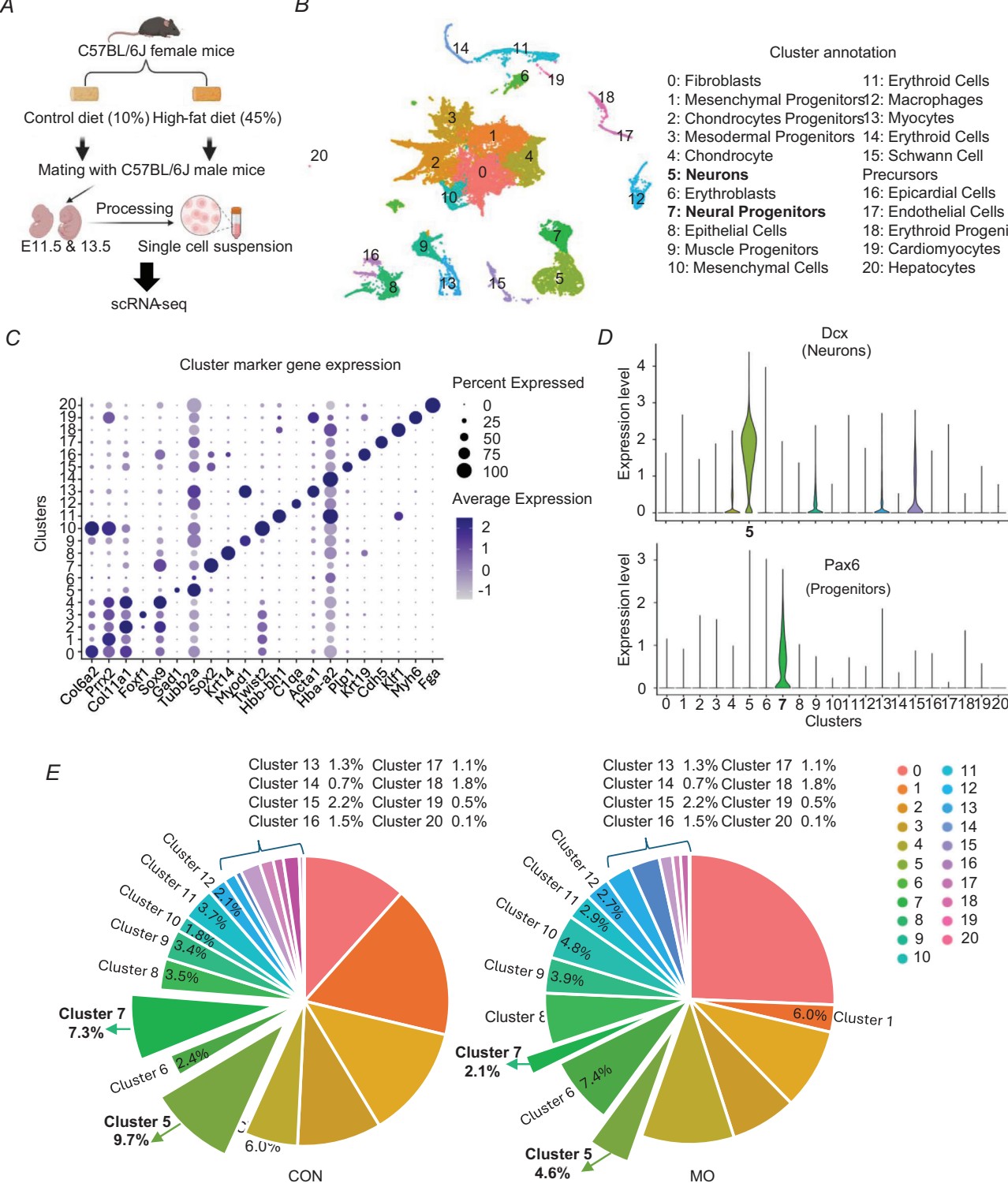

**Figure 1. Single-cell transcriptomic profiling of mouse embryos on embryonic days 11.5 and 13.5**
*A*, experimental design showing the diet-induced maternal obesity (MO) model and embryo collection time points. *B*, UMAP (uniform manifold approximation and projection) visualization of integrated single-cell clusters from E11.5 and E13.5 embryos. *C*, canonical marker gene expression across identified cell clusters. *D*, violin plots showing the expression of neural marker genes. *E*, comparison of cell proportions between CON (control) and MO groups.

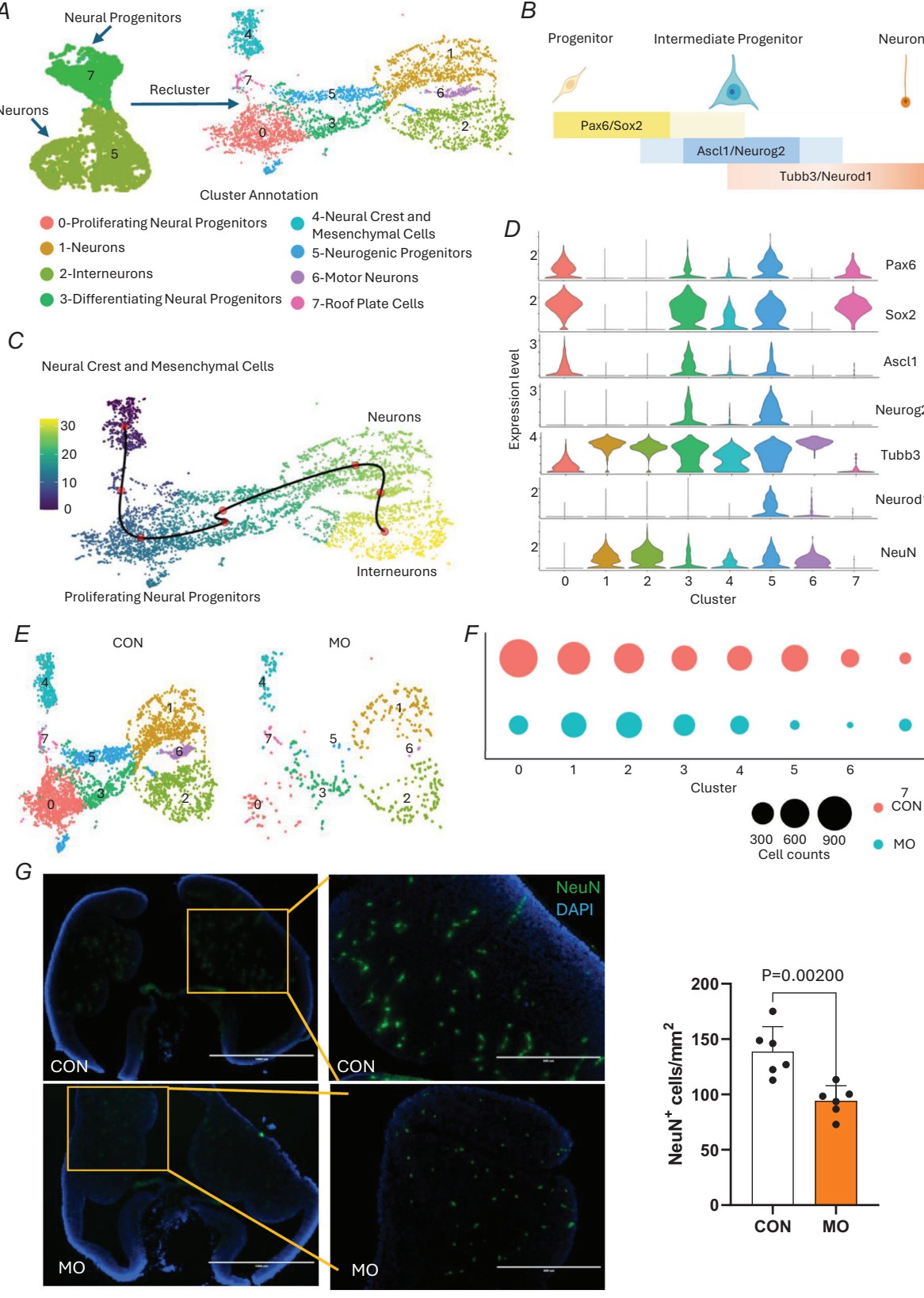

**Figure 2. Subclustering analysis of neural populations**

*A*, UMAP (uniform manifold approximation and projection) visualization of re-clustered neural populations from the integrated dataset. *B*, schematic diagram shows the progression of embryonic neurogenesis from progenitor cells

to mature neurons. *C*, pseudotime trajectory analysis of neural cell differentiation. *D*, differential gene expression pattern across neural clusters. *E*, UMAP visualization comparing neural cluster distribution between control (CON) and maternal obesity (MO) groups. *F*, comparison of neural subpopulation cell numbers between CON and MO groups. *G*, representative immunofluorescence images of E13.5 embryonic brain from CON and MO stained for NeuN (green), a marker of postmitotic neurons, with DAPI (blue) labelling nuclei. Low-magnification images show the overall brain morphology and the region selected for analysis (boxed). High-magnification views correspond to the ganglionic eminence in the ventral telencephalon. Quantification shows a reduced density of NeuN$^+$ neurons in the MO group compared with the CON group. Data are presented as mean ± SD, and each dot represents one litter (*n* = 6). *P*-value in CON *versus* MO using unpaired Student's *t* test. Scale bar = 1000 μm (overview) and 400 μm (high magnification).

(expressing *Pax6/Sox2*) through intermediate progenitors (expressing *Ascl1/Neurog2*) to mature neurons (expressing *Tubb3/Neurod1*). MO led to a substantial reduction in both neurons (cluster 5: 9.7% CON *vs.* 4.6% MO) and neural progenitors (cluster 7: 7.3% CON *vs.* 2.1% MO), indicating an overall decrease in neural populations in the MO group (Fig. 1*E*). Furthermore when CON and MO were compared, we observed sparse distributions and fewer cell counts among the neuronal subtypes in MO (Fig. 2*E* and *F*). We then performed immunofluorescence staining of E13.5 brain using NeuN, a marker of post-mitotic neurons. A significant reduction in neuronal populations was observed in MO compared with CON, providing direct *in vivo* evidence that MO impairs neuro-genesis in the embryonic brain (Fig. 2*G*). Together with the overall decline in neurogenic populations, changes in distribution imply that MO not only lowers the overall number of neural cells but also disrupts the differentiation of neurogenic cells.

**Maternal obesity disrupts neurogenic marker expression.**
We examined the expression of key neurogenic markers in neural clusters using scRNA-seq. Violin plots show the expression levels of *Neurod1*, *Neurog2* and *Ascl1* in the CON and MO groups. Compared to the CON group a significant downregulation of *Neurod1*, *Neurog2* and *Ascl1* was observed in the MO group (Fig. 3*A*). To further examine the findings from single-cell trans-criptome analysis, we performed RT-qPCR to analyse the expression of these neurogenic markers in E13.5 brain samples from the CON and MO groups. Similarly the MO E13.5 brains exhibited significant down-regulation of *Neurod1*, *Neurog2* and *Ascl1* compared to the CON group (Fig. 3*A*). These results collectively confirm that MO impairs the neurogenic process, as evidenced by the downregulation of *Neurod1*, *Neurog2* and *Ascl1* in both scRNA-seq and RT-qPCR analyses. GO enrichment analysis of downregulated genes in the MO revealed enrichment of neurogenesis and nervous

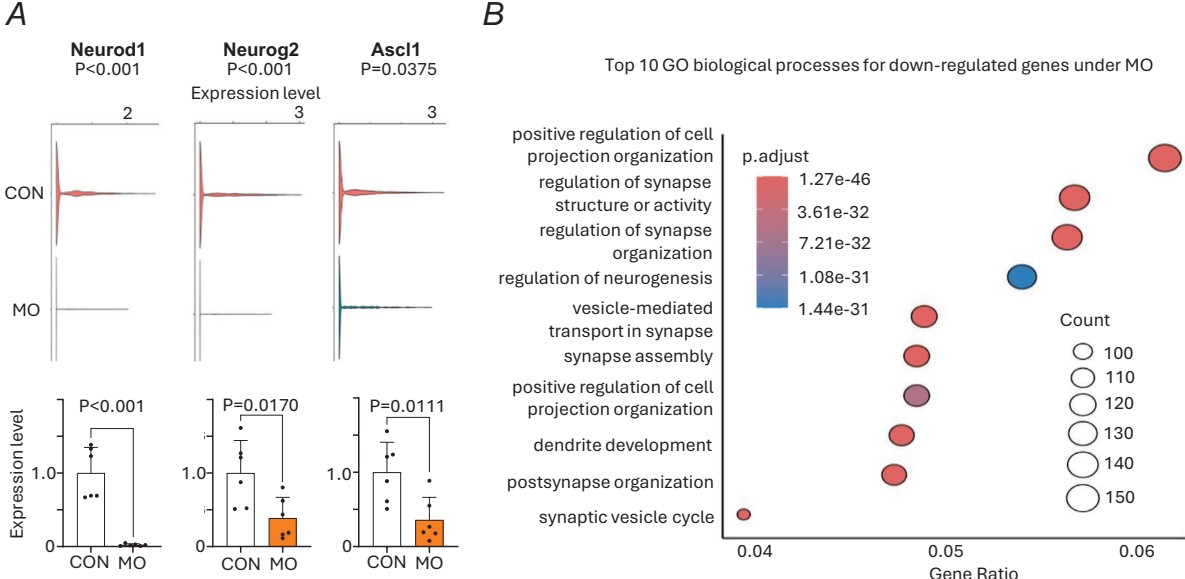

**Figure 3. Effect of MO (maternal obesity) on neurogenic gene expression**
*A*, expression of key neurogenic transcription factors (*Neurod1*, *Neurog2*, *Ascl1*) in neural clusters using scRNA-seq (shown in violin plots); RT-qPCR validation of neurogenic marker expression in E13.5 brain samples from CON (control) and MO groups (shown in bar plots). *B*, Gene Ontology enrichment analysis of downregulated genes in MO, highlighting impaired neurodevelopmental pathways. Data are presented as mean ± SD, and each dot represents one litter (*n* = 6). *P*-value in CON *versus* MO using unpaired Student's *t* test.

system development pathways, showing impaired developmental processes (Fig. 3*B*). Genes involved in synaptic organization and dendrite development were also affected, suggesting that MO impacts not only neuronal differentiation but neuronal connectivity.

**MO induces an inflammatory response in embryos.** To investigate the inflammatory response in embryonic tissue due to MO, we first identified immune cell populations using canonical markers and revealed their predominant expression in cluster 12, identifying this cluster as the immune cell population (Fig. 4*A*). Further examination of inflammatory markers within this population revealed differential expression patterns between the MO and CON groups (Fig. 4*B* and *D*). Several inflammatory mediators, including TNF and CXCL2, and the immediate early genes *FOS*, *JUN* and *JUND* were significantly higher in the MO group, whereas CCL2 exhibited trending but non-significant increases. These findings suggest that MO induces an inflammatory state in the embryo, particularly evidenced by the upregulation of key inflammatory mediators. The significant elevation of TNF and CXCL2, along with the robust increase in immediate early genes, indicates activation of pro-inflammatory pathways that may provide a mechanistic link between MO and impaired neurogenesis. To validate we analysed the protein levels of TNF-$\alpha$ and JUN, a core component of the AP-1 transcription factor complex. Both TNF-$\alpha$ and JUN protein contents were significantly increased in embryos from obese mothers compared to CONs. These protein-level changes are consistent with our scRNA-seq data, which showed elevated expression of inflammatory cytokines and AP-1 pathway genes under MO (Fig. 4*C* and *E*). This inflammatory environment could be a key mechanism by which MO impacts embryonic neural development.

**MO alters chromatin accessibility in inflammatory gene regulatory regions.** We performed scATAC-seq on E13.5 embryos from CON and MO to explore the chromatin accessibility profiles (Fig. 5*A*). After E13.5 samples were collected and scATAC-seq analysis was performed, we utilized scRNA-seq data to predict and annotate the scATAC-seq clusters for better cell population characterization (Fig. 5*B*). Our analysis of motif activity focused on cluster 12 in the predicted scATAC-seq data, which corresponded to an immune cell population based on scRNA-seq clustering. Motif activity analysis revealed significant upregulation of AP-1 family transcription factors in MO, with average difference scores (avg_diff) for FOS (4.73), FOS::JUN (4.59) and FOSL1 (4.51) (Fig. 5*C*). These findings suggest enhanced activity of AP-1 family transcription factors in immune cells under MO, suggesting activation of inflammatory pathways.

We first examined accessibility at *Tnf* and *Cd68* promoter regions, which were increased in MO compared to CON across all clusters (Fig. 5*D*). The chromatin accessibility changes observed at these loci suggest enhanced transcriptional regulation of *Tnf*, a key pro-inflammatory cytokine, and *Cd68*, a macrophage marker. We further analysed the AP-1 binding sites within the *Tnf* and *Cd68* promoter regions. In the *Tnf* two AP-1 binding sites, which are JUN and JUND, were found; in the *Cd68* multiple AP-1 binding sites, including JUN, FOS::JUNB, FOSB::JUNB, FOSL2::JUNB, FOSL2::JUND, FOS::JUN, FOSL2::JUN and FOS::JUND, were found. The accessibility was higher in MO together with AP-1 binding sites on the *Tnf* and *Cd68* promoter regions and the increased AP-1 motif activity, indicating inflammatory response due to MO conditions.

To validate chromatin accessibility we performed ChIP-qPCR. Using an antibody against c-JUN we observed significantly greater enrichment of c-JUN binding at the *Tnf* promoter in E13.5 embryos from the MO group compared to the CON group. These results are consistent with our scATAC-seq findings, which revealed enhanced AP-1 motif activity and increased chromatin accessibility at the *Tnf* locus under MO (Fig. 5*E*).

***In vitro* validation of inflammatory impact on neurogenesis.** To investigate the impact of inflammatory cytokines on neurogenesis-related gene expression, neural cell cultures were treated with varying concentrations of TNF-$\alpha$ (0, 10, 30 and 100 ng/mL) (Fig. 6). RT-qPCR analysis revealed that treatment with 30 and 100 ng/mL TNF-$\alpha$ resulted in significantly reduced expression levels of *Neurog2* and *Neurod1* compared to the untreated CON (0 ng/mL). However lower concentrations (10 ng/mL) did not significantly affect the expression of these genes. Additionally TNF-$\alpha$ stimulation exhibited no significant effect on the expression of *Ascl1*. These findings suggest a dose-dependent inhibitory effect of TNF-$\alpha$ on neurogenesis-related gene expression in neural cells.

## Discussion

Obesity rate is increasing worldwide, incurring huge medical costs (Patel et al., 2015; Zhang et al., 2024). MO is even more concerning as it affects not only mothers but also their offspring. Increasing research work on both humans and animals suggests that MO leads to detrimental impacts on cognitive function, including depression, anxiety, ASD and ADHD in offspring (Contu & Hawkes, 2017; Edlow, 2017; van der Burg et al., 2016). Human cohort studies have demonstrated that higher pre-pregnancy maternal body mass index (BMI) is consistently associated with lower scores in tests

assessing visual-motor abilities, with inflammation during pregnancy identified as a key mediator of these cognitive deficits (Monthe-Dreze et al., 2019). However the underlying mechanisms remain poorly defined (Leong, 2018; O'Reilly & Reynolds, 2013).

MO is known to induce chronic, low-grade inflammation, which has been linked to the development of disorders in the offspring (Pantham et al., 2015). The sequential expression of transcription factors *Neurog2*, *Ascl1* and *Neurod1* directs proper neuronal

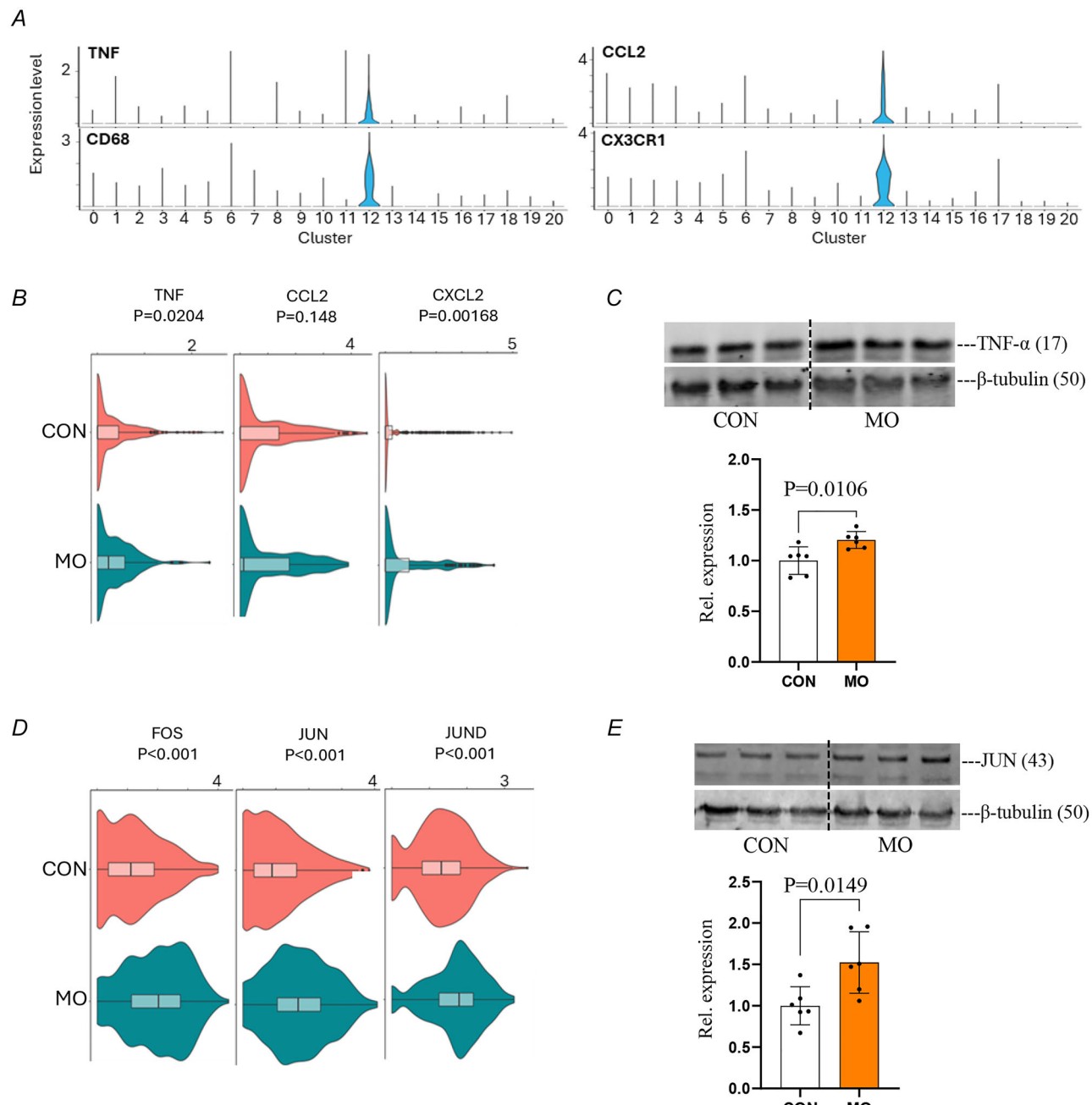

**Figure 4. Inflammatory gene expression analysis in CON (control) and MO (maternal obesity) groups using scRNA-seq**
*A*, violin plots showing the expression of inflammatory markers. *B*, inflammatory gene expression between CON and MO groups in immune cell cluster. *C*, cropped Western blots of TNF-α (tumour necrosis factor-α) (β-tubulin as loading control) from CON and MO. *D*, AP-1 (activator protein 1) transcription factor component gene expression between CON and MO groups in immune cell cluster. *E*, cropped Western blots of JUN (β-tubulin as loading control) from CON and MO. Data are presented as mean ± SD, and each dot represents one litter (*n* = 6). *P*-value in CON *versus* MO using unpaired Student's *t* test.

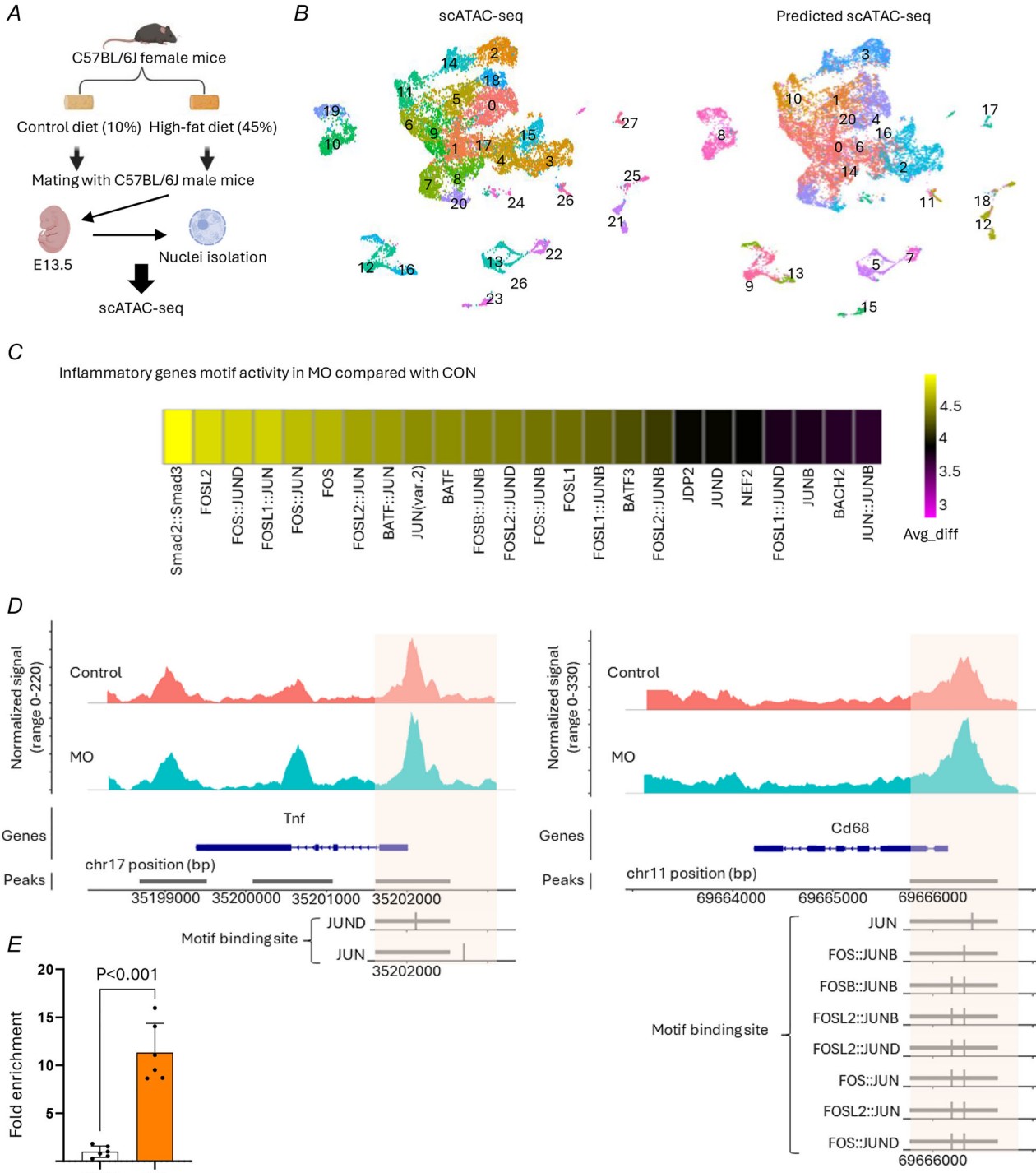

**Figure 5. scATAC-seq in E13.5 embryos**
*A*, experimental workflow for single-cell ATAC sequencing. *B*, UMAP (uniform manifold approximation and projection) visualization of scATAC-seq clusters with predicted cell-type annotations from integrated scRNA-seq. *C*, differential motif activity analysis showing upregulation of AP-1 (activator protein 1) family transcription factors in MO (maternal obesity). *D*, chromatin accessibility at *Tnf* and *Cd68* gene loci in CON (control) and MO. AP-1 binding sites were identified, with RelScores ranging from 0.90 to 0.95. *E*, increased c-JUN binding at the *Tnf* promoter in E13.5 embryos from MO. ChIP (chromatin immunoprecipitation)-qPCR showing enrichment of c-JUN binding at the *Tnf* promoter region in E13.5 embryonic tissues from CON and MO groups. Data are presented as mean ± SD, and each dot represents one litter (*n* = 6). *P*-value in CON *versus* MO using unpaired Student's *t* test.

development and differentiation (Roybon et al., 2010). Our findings demonstrate significant downregulation of neurogenic markers in offspring in this cascade under MO conditions. Specifically our scRNA-seq and RT-qPCR analyses revealed significant downregulation of *Neurod1*, *Neurog2* and *Ascl1* in MO offspring, suggesting altered neuronal differentiation potential. The reduction in *Neurod1* expression is particularly concerning as it serves as a master regulator of neuronal differentiation and maturation (Pataskar et al., 2016). The overall disruption of these transcription factors aligns with the reduced proportion of neurons and neural stem cells in MO observed in our cluster analysis. GO analysis showed the downregulation of genes in neurogenesis and nervous system development in the MO group. These findings are consistent with previous studies showing that impairment in these key developmental regulators can have lasting impacts on neural development and function (Dennis et al., 2019; Roybon et al., 2009).

Our scRNA-seq analyses revealed the inflammatory mechanisms underlying these neurogenic disruptions. We identified increased expression of pro-inflammatory cytokines such as *Tnf* and *Cxcl2* in MO, demonstrating an active inflammatory response. Further analysis through scATAC-seq confirmed increased motif activity for FOS, FOS::JUN and FOSL1, suggesting the enhanced inflammatory response was likely responsible for the impairment in neurogenesis. These immediate early genes, which form the AP-1 transcription factor complex, are known to be rapidly activated in response to inflammatory stimuli (Vukic et al., 2009). The increased expression of these factors is particularly significant as they serve as a crucial link between inflammation and altered gene expression patterns (Qiao et al., 2016). Sustained activation of AP-1 factors can promote the expression of pro-inflammatory cytokines while simultaneously interfering with normal developmental programming (Vukic et al., 2009; Wei et al., 2019). In neural stem cells excessive AP-1 activity has been shown to disrupt the expression of key neurogenic factors and alter cell fate decisions (Wei et al., 2019). The concurrent elevation of these factors with reduced neurogenic gene expression in our study suggests that AP-1-mediated inflammatory response might be a key mechanism by which MO impairs fetal neurogenesis. In addition our scATAC-seq analysis revealed significantly increased chromatin accessibility at *Tnf* and *Cd68* gene loci in MO compared to CON. This enhanced accessibility at the *Tnf* promoter region supports the observed upregulation of this key pro-inflammatory cytokine, whereas increased accessibility at the *Cd68* locus indicates enhanced fetal macrophage activation in response to the maternal inflammatory environment. The fetal macrophages are detectable in embryonic tissues after E12.5 (Wu & Hirschi, 2020). The elevated AP-1 motif activity in MO and the presence of AP-1 binding sites in the *Tnf* and *Cd68* promoter regions suggest that AP-1 mediates inflammatory response in MO embryos. These findings align with previous research demonstrating that maternal immune activation creates a pro-inflammatory environment that can cross the placental barrier and increase the number of activated choroid plexus macrophages, affecting fetal development at E14.5 (Cui et al., 2020; Goyal et al., 2019; Kwon et al., 2022). The increased expression of cytokines by MO creates an inflammatory environment that impairs proper neurogenic programming.

To establish a direct link between inflammation and impaired neurogenesis, our *in vitro* study demonstrated that TNF-α stimulation during neural stem cell differentiation significantly reduced the expression of key neurogenesis markers, *Neurod1* and *Neurog2*, which

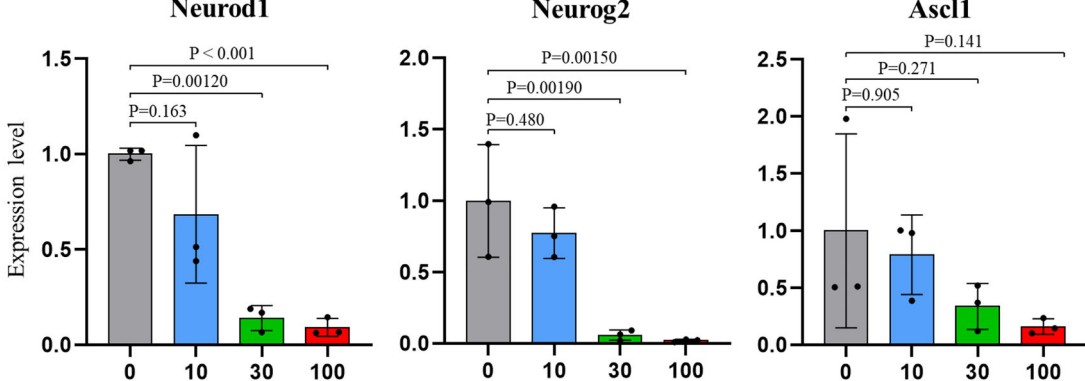

**Figure 6. Effect of TNF-α (tumour necrosis factor-α) on *Neurod1*, *Neurog2* and *Ascl1* expression in neural cell cultures**
Neural cells were treated with TNF-α at 0, 10, 30 and 100 ng/mL for 48 hours. RT-qPCR was performed to measure the expression levels of *Neurod1*, *Neurog2* and *Ascl1*. Data are presented as mean ± SD, and each dot represents one independent experiment (*n* = 3). *P*-value in 10, 30, 100 ng/mL treatments *versus* 0 using one-way ANOVA.

are crucial for neuronal differentiation (Aslanpour et al., 2020). This data suggests that inflammation disrupts neuronal differentiation, consistent with previous studies showing that TNF-$\alpha$ inhibits the proliferation of NPCs and suppresses neurogenic differentiation (Ben-Hur et al., 2003; Keohane et al., 2010).

In conclusion our study provides evidence that MO significantly impairs fetal neurogenesis by inducing inflammatory response. Using a combination of scRNA-seq, scATAC-seq and *in vivo* and *in vitro* validation, we demonstrated that MO leads to reduced expression of key neurogenic factors and an increase in inflammatory markers, supported by changes in chromatin accessibility at *Tnf* and Cd68 gene loci. The reduction in neuronal and neural stem cell clusters, together with the increased expression of pro-inflammatory genes, suggests that MO accelerates inflammation that interferes with regular neuro-developmental programming. These molecular alterations may result in an increased risk of neuro-developmental disorders observed in the offspring of obese mothers. Our findings highlight the importance of maintaining a healthy weight during pregnancy and suggest that targeting inflammatory pathways could be a potential therapeutic strategy to mitigate the adverse effects of MO on fetal brain development. Future studies should focus on developing interventions that can protect fetal neurodevelopment from the detrimental impacts of MO-induced inflammation.

The present study focuses on early embryonic neurogenesis, and E11.5–E13.5 represents a critical developmental window during which neural progenitor proliferation, proneural commitment and early neuro-nal differentiation are highly active. In addition early neural induction remains active at the developing end of the neural tube. Thus, this developmental period enables us to investigate how MO affects early neurogenic programming. By integrating single-cell transcriptomic and epigenomic analyses with *in vivo* and *in vitro* validation at these early stages, our study captures primary molecular and cellular disruptions in neurogenesis induced by MO. Indeed future studies exploring later differentiation and maturation stages (E17.5–postnatal) will be important for defining the full developmental trajectory. Nevertheless our focused analysis of early neurogenesis provides valuable mechanistic insight into the inflammatory and transcriptional programmes that impair the early neural development due to MO.

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

## Additional information

### Data availability statement

The data for this study are available in the Gene Expression Omnibus (GEO), under accession number GSE278631, https://www.ncbi.nlm.nih.gov/geo/query/acc.cgi?acc=GSE278631.

### Competing interests

The authors declare no known competing financial interests or personal relationships.

### Author contributions

L.-W.C. and M.D. designed and planned the work. L.-W.C., M.N.H., Y.G., Z.K., S.I., X.L., C.S. and J.M.A. performed the animal experiments. L.-W.C. and M.N.H. participated in data analyses. L.-W.C., M.N.H. and M.D. participated in data interpretation. M.D. and M.Z. supervised the study. L.-W.C. wrote the original manuscript with inputs from M.D. L.-W.C. and M.D. reviewed and edited the manuscript.

### Funding

This research work was supported by the National Institutes of Health Grant R01HD067449.

### Keywords

embryo, inflammation, maternal obesity, neurogenesis, neurogenic transcription factors

## Supporting information

Additional supporting information can be found online in the Supporting Information section at the end of the HTML view of the article. Supporting information files available:

**Peer Review History**
**Figure A1**
**Figure A2**

