## [Peer Review History · The Journal of Physiology]

Maternal Obesity Induces AP-1-Mediated Inflammatory Response to Impair Embryonic Neurogenesis

Li Wei Chen, Md Nazmul Hossain, Yao Gao, Zhongyun Kou, Sharmeen Islam, xinrui li, Chaeyoung Shin, Jeanene Marie de Avila, Meijun Zhu, and Min Du

DOI: 10.1113/JP289326

Corresponding author(s): Min Du (min.du@wsu.edu)

Review Timeline:

Submission Date:	23-May-2025
Editorial Decision:	15-Jul-2025
Revision Received:	06-Jan-2026
Editorial Decision:	16-Jan-2026
Revision Received:	22-Jan-2026
Accepted:	24-Feb-2026

Senior Editor: Laura Bennet

Reviewing Editor: Justin Dean

Transaction Report:

Dear Dr Du,

Re: JP-RP-2025-289326 **"Maternal Obesity Induces AP-1-Mediated Inflammatory Response to Impair Embryonic Neurogenesis"** by Li Wei Chen, Md Nazmul Hossain, Yao Gao, Zhongyun Kou, Sharmeen Islam, xinrui li, Chaeyoung Shin, Jeanene Marie de Avila, Meijun Zhu, and Min Du

Thank you for submitting your manuscript to The Journal of Physiology. It has been assessed by a Reviewing Editor and by 2 expert referees and we are pleased to tell you that it is potentially acceptable for publication following satisfactory major revision.

LANGUAGE EDITING AND SUPPORT FOR PUBLICATION: If you would like help with English language editing, or other article preparation support, Wiley Editing Services offers expert help, including English Language Editing, as well as translation, manuscript formatting, and figure formatting at www.wileyauthors.com/eoo/preparation. You can also find resources for Preparing Your Article for general guidance about writing and preparing your manuscript at www.wileyauthors.com/eoo/prepresources.

REVISION CHECKLIST:

We look forward to receiving your revised submission.

Yours sincerely,

Laura Bennet
Senior Editor
The Journal of Physiology

REQUIRED ITEMS

- The contact information for the person responsible for 'Research Governance' at your institution needs to be provided. This includes their name and an institutional email address. Please ensure the contact is not an author on this paper and provide an alternate contact if necessary, or confirm in the submission form that the author whose email was provided has sole responsibility for research governance. This is the person who is responsible for regulations, principles and standards of good practice in research carried out at the institution, for instance the ethical treatment of animals, the keeping of proper experimental records or the reporting of results.
- The reference list must be in alphabetical order, rather than numbered, to comply with our Journal format.
- Your manuscript must include a complete Additional Information section, including competing interests; funding; author contributions and acknowledgements.
- Please upload separate high-quality figure files via the submission form.
- Please ensure that any tables are editable and in Word format, and wherever possible, embedded in the article file itself.
- Please ensure that the Article File you upload is a Word file.
- A Data Availability Statement is required for all papers reporting original data. This must be in the Additional Information section of the manuscript itself. It must have the paragraph heading 'Data Availability Statement'. All data supporting the results in the paper must be either: in the paper itself; uploaded as Supporting Information for Online Publication; or archived in an appropriate public repository. The statement needs to describe the availability or the absence of shared data. Authors must include in their statement: a link to the repository they have used, or a statement that it is available as Supporting Information; reference the data in the appropriate sections(s) of their manuscript; and cite the data they have shared in the References section. Whenever possible, the scripts and other artefacts used to generate the analyses presented in the paper should also be publicly archived. If sharing data compromises ethical standards or legal requirements then authors are not expected to share it, but must note this in their statement. For more information, see our Statistics Policy.

EDITOR COMMENTS

Reviewing Editor:

Comments to the Author:

Thank you for your submission. Both reviewers were generally supportive of the manuscript, but suggest a number of major revisions and additional experiments to improve the manuscript and its potential impact. These include, but are not limited to, clarifying what embryonic cells were used for single-cell RNA-seq. (at a minimum isolating brain-derived cells for single-cell analysis) and perform relevant supporting protein and histological analyses of the brain tissues-without these the conclusion are not supported; add additional pathway analyses and RNA/western blot analyses for inflammatory genes; considerations of improved temporal resolution in RNA-seq data; and add further details on the maternal metabolic phenotype and other animal data. Please also add information on euthanasia.

REFeree COMMENTS

Referee #1:

In this study, the authors utilized a maternal obesity (MO) model and conducted a comprehensive single-cell RNA sequencing analysis of embryonic cells collected at E11.5 and E13.5, comparing them to controls (standard diet). The authors claim that single-cell RNA sequencing revealed reduced proportions of neurons and neural progenitors in embryos from obese mothers, indicating compromised neurogenesis. They also described that Gene Ontology analysis supported this finding by showing down-regulation of pathways related to neurogenesis, nervous system development, and synaptic organization.

However, the manuscript does not clearly specify which embryonic cells were used for the single-cell RNA-seq analysis. It appears that whole embryos may have been used, but if that is the case, it is not possible to determine which regions of the nervous system are affected. In the Introduction and Discussion, the authors mention that MO has been implicated in increased risk of neurodevelopmental disorders such as ASD and ADHD, suggesting potential effects on the brain. Yet, the actual impact of MO on the embryonic brain remains unclear.

Given this context, if the authors wish to draw conclusions about neural populations, they should at minimum isolate brain-derived cells for single-cell analysis. Moreover, histological examination of the brain is essential. While the study includes in vitro experiments using neural cells, in vivo validation is equally important. Alternatively, if previous studies have addressed this point, appropriate references should be cited at relevant points in the manuscript.

Referee #2:

In this manuscript entitled "Maternal Obesity Induces AP-1-Mediated Inflammatory Response to Impair Embryonic Neurogenesis," Chen et al. explore the impact of maternal obesity (MO) on early embryonic brain development. By employing single-cell transcriptomic (scRNA-seq) and epigenomic (scATAC-seq) technologies, they tried to uncover how a high-fat maternal diet alters neural lineage composition, neurogenic gene expression, and chromatin accessibility in the developing brain. The findings are of relevance to our understanding of developmental origins of neurodevelopmental disorders, highlighting inflammation-driven impairment in neurogenesis as a plausible mechanistic link between maternal metabolic state and offspring brain dysfunction.

The experimental design is conceptually strong. The study models maternal obesity by exposing pregnant females to a sustained high-fat diet prior to and during gestation. The integration of scRNA-seq and scATAC-seq provides powerful, high-resolution insights into both transcriptional and regulatory dynamics of embryonic neural development. A key strength is the identification of disrupted neural progenitor-to-neuron differentiation trajectories in MO embryos, demonstrated by reduction in Pax6⁺/Sox2⁺ neural progenitors and Ascl1⁺/Neurog2⁺ intermediate progenitors, and a corresponding decrease in Neurod1⁺/Tubb3⁺ neurons. The study further strengthens its conclusions by validating the suppression of neurogenic transcription factors (Neurod1, Neurog2, Ascl1) through RT-qPCR and connecting these alterations to inflammatory signalling via the upregulation of Tnf, Fos, Jun, and Cxcl2.

The use of scATAC-seq adds depth by identifying increased chromatin accessibility at AP-1 transcription factor motifs near the Tnf and Cd68 genes, consistent with an inflammatory transcriptional shift. Importantly, TNF- α treatment of neurogenic cells in vitro led to reduced Neurod1 and Neurog2 expression, establishing a functional link between inflammation and impaired neurogenesis.

Major concerns

Despite these strengths, there are key limitations that must be addressed before considering it for publication.

The major issue is that the study lacks protein-level validation for all the experiments, which is needed to verify the claims made in the study.

Figure 2B: The stem cell differentiation markers should be analyzed by immunohistochemistry for control and GDM samples of the same age and presented.

Figure 3: The downregulation of Neurod1, Neurog2, and Ascl1 is shown at the mRNA level; however, confirmation by immunohistochemistry and Western blot is essential for publication. Protein-level evidence would also support the claim that MO-induced inflammation impacts neurogenic fate at the translational level. Add immunohistochemistry and Western blot validation for NEUROD1, NEUROG2, and ASCL1 in embryonic brains (immunostaining or Western blot).

It is also beneficial to include some of the other known components, such as Neural Stem/Progenitor Cell Maintenance, Proneural Commitment Regulators, Neuronal Differentiation and Maturation, and their modulators.

Figure 4: Please include qRT-PCR and western blots for the inflammatory genes explained.

Fig.5: Verify the chromatin acceptability changes by Chip analysis

Fig. 6 verifies the results by Western blot

Another major issue is that the maternal metabolic phenotype is not sufficiently characterized. While the maternal diet is clearly defined, there is no data on maternal insulin, glucose, lipid profile including, HDL, LDL, triglycerides, total cholesterol and systemic inflammatory markers (e.g., TNF- α , IL-6, CRP), which are necessary to correlate maternal status with fetal outcomes and establish the broader physiological relevance.

Additionally, the weight, age, lipid, and glucose levels of the male animals used in the experiments are also required, as variations in their weights and age have been shown to influence the metabolic status of the offspring.

Another major limitation is that the study samples embryos only at E11.5 and E13.5, which limits the temporal resolution. Key developmental events such as early neural induction (around E9.5) and terminal differentiation or synaptic maturation (E17.5-P0) are not captured. A more complete developmental trajectory would greatly strengthen the conclusions.

Minor concerns:

The raw data should be submitted, and the link is needed to verify the data which is missing.

Explain the neural stem cell culture media and the differentiation media clearly. The components, along with their concentrations, will facilitate the repetition of these experiments.

Overall, the study is well-conceived and presents compelling preliminary evidence that maternal obesity impairs early neurogenesis through inflammation-driven transcriptional and epigenetic reprogramming. However, before submission, the concerns raised needed to be addressed.

END OF COMMENTS

EDITOR COMMENTS

Reviewing Editor:

Comments to the Author:

Thank you for your submission. Both reviewers were generally supportive of the manuscript, but suggest a number of major revisions and additional experiments to improve the manuscript and its potential impact. These include, but are not limited to, clarifying what embryonic cells were used for single-cell RNA-seq. (at a minimum isolating brain-derived cells for single-cell analysis) and perform relevant supporting protein and histological analyses of the brain tissues-without these the conclusion are not supported; add additional pathway analyses and RNA/western blot analyses for inflammatory genes; considerations of improved temporal resolution in RNA-deq data; and add further details on the maternal metabolic phenotype and other animal data. Please also add information on euthanasia.

Response:

We appreciate the editor for the careful evaluation of our manuscript and for the comments on improving this article's impact. We have performed substantial additional experiments to address the concerns raised by reviewers, including histological examination of embryonic brain, western blot, metabolic characterization, and validation of AP-1 mediated regulation using ChIP-qPCR.

Also, we have added information regarding animal euthanasia procedures to the Materials and Methods section (Animal treatments).

Revised Text:

Pregnant mice were anesthetized by carbon dioxide inhalation and euthanized by cervical dislocation at E11.5 and E13.5.

REFeree COMMENTS

Referee #1:

In this study, the authors utilized a maternal obesity (MO) model and conducted a comprehensive single-cell RNA sequencing analysis of embryonic cells collected at E11.5 and E13.5, comparing them to controls (standard diet). The authors claim that single-cell RNA sequencing revealed reduced proportions of neurons and neural progenitors in embryos from obese mothers, indicating compromised neurogenesis. They also described that Gene Ontology analysis supported this finding by showing down-regulation of pathways related to neurogenesis, nervous system development, and synaptic organization.

Major comments:

1. However, the manuscript does not clearly specify which embryonic cells were used for the single-cell RNA-seq analysis. It appears that whole embryos may have been used, but if that is the case, it is not possible to determine which regions of the nervous system are affected. In the Introduction and Discussion, the authors mention that MO has been implicated in increased risk of neurodevelopmental disorders such as ASD and ADHD, suggesting potential effects on the brain. Yet, the actual impact of MO on the embryonic brain remains unclear.

Response:

Thank you for the comments. We have now clarified that embryos used for single-cell RNA sequencing were processed at embryonic days 11.5 and 13.5.

At these early developmental stages, the nervous system is mainly present as neural tubes and associated neural crest. The trunk region of the mouse embryo contains key neurogenic structures, including the spinal neural tube and neural crest-derived populations. The spinal neural tube at E11.5–E13.5 remains highly proliferative and actively generates neurons and neural progenitors, and therefore constitutes a relevant neurogenic compartment for assessing early neurodevelopmental processes [1]. This approach allowed us to investigate neural progenitor and neuronal populations while simultaneously capturing inflammatory and immune cell populations within the embryonic environment.

To directly address the actual impact of MO on the region of primitive embryonic brain, we have now performed *in vivo* validation using immunofluorescence staining for neuronal markers.

2. Given this context, if the authors wish to draw conclusions about neural populations, they should at minimum isolate brain-derived cells for single-cell analysis. Moreover,

histological examination of the brain is essential. While the study includes in vitro experiments using neural cells, in vivo validation is equally important. Alternatively, if previous studies have addressed this point, appropriate references should be cited at relevant points in the manuscript.

Response:

Thanks! By using embryos for single-cell sequencing, specific cell types can be virtually sorted based on their specific transcription profiles. . In this study, we virtually isolate neural populations for detailed assessment of changes specific to neurogenic cells. In addition, whole embryo sequencing also allows us to simultaneously assess inflammation and other changes in associated cells, which contribute to the adverse embryonic environment and thus impairment of neurogenesis due to maternal obesity. The primary objective is to examine how maternal obesity affects neurogenic commitment and early neuronal differentiation, which is largely independent of specific neural tube or brain regions.

Additionally, per your suggestion, we performed histological examination of the embryonic brain using immunofluorescence staining for neuronal markers to validate conclusions regarding embryonic neurogenesis. Data show a significant reduction in neuronal populations in MO compared with CON, providing direct in vivo evidence that maternal obesity impairs neurogenesis in the embryonic brain.

Figure1. **MO reduces neuronal populations in the E13.5 brain in vivo.**

Representative immunofluorescence images of E13.5 mouse embryonic brain from CON and MO stained for NeuN (green), a marker of post-mitotic neurons, with DAPI (blue) labeling nuclei. Data are presented as mean \pm SEM (n = 6). * $P < 0.05$. Scale bar = 400 μ m.

Referee #2:

In this manuscript entitled "Maternal Obesity Induces AP-1-Mediated Inflammatory Response to Impair Embryonic Neurogenesis," Chem et al. explore the impact of maternal obesity (MO) on early embryonic brain development. By employing single-cell transcriptomic (scRNA-seq) and epigenomic (scATAC-seq) technologies, they tried to uncover how a high-fat maternal diet alters neural lineage composition, neurogenic gene expression, and chromatin accessibility in the developing brain. The findings are of relevance to our understanding of developmental origins of neurodevelopmental disorders, highlighting inflammation-driven impairment in neurogenesis as a plausible mechanistic link between maternal metabolic state and offspring brain dysfunction.

The experimental design is conceptually strong. The study models maternal obesity by exposing pregnant females to a sustained high-fat diet prior to and during gestation. The integration of scRNA-seq and scATAC-seq provides powerful, high-resolution insights into both transcriptional and regulatory dynamics of embryonic neural development. A key strength is the identification of disrupted neural progenitor-to-neuron differentiation trajectories in MO embryos, demonstrated by reduction in Pax6⁺/Sox2⁺ neural progenitors and Ascl1⁺/Neurog2⁺ intermediate progenitors, and a corresponding decrease in Neurod1⁺/Tubb3⁺ neurons. The study further strengthens its conclusions by validating the suppression of neurogenic transcription factors (Neurod1, Neurog2, Ascl1) through RT-qPCR and connecting these alterations to inflammatory signalling via the upregulation of Tnf, Fos, Jun, and Cxcl2.

The use of scATAC-seq adds depth by identifying increased chromatin accessibility at AP-1 transcription factor motifs near the Tnf and Cd68 genes, consistent with an inflammatory transcriptional shift. Importantly, TNF- α treatment of neurogenic cells in vitro led to reduced Neurod1 and Neurog2 expression, establishing a functional link between inflammation and impaired neurogenesis.

Major concerns

1. Despite these strengths, there are key limitations that must be addressed before considering it for publication. The major issue is that the study lacks protein-level validation for all the experiments, which is needed to verify the claims made in the study.

Response:

Thank you for your thoughtful comments. We have now performed immunohistochemical staining using a neuron marker for validating the reduction in neural population, and western blot to assess the protein expression of key neurogenic and inflammatory markers implicated in our study. Maternal metabolic profile, including maternal insulin, glucose, lipid profile and TNF-alpha were also analyzed and added.

2. Figure 2B: The stem cell differentiation markers should be analyzed by immunohistochemistry for control and GDM samples of the same age and presented.

Response:

Figure 2B is a schematic summary illustrating the well-established progression of embryonic neurogenesis from neural progenitors to mature neurons. The differentiation stages and marker expression shown in this schematic are based on canonical neurodevelopmental frameworks. The expression patterns of these markers are supported by our single-cell RNA sequencing data, which show distinct, stage-specific expression of Pax6/Sox2, Ascl1/Neurog2, and Neurod1/Tubb3 across neural subclusters (Figure 2D).

Specifically, early neural progenitor identity is characterized by expression of Pax6 and Sox2, which are known to regulate radial glial cell proliferation and neurogenic competence. Pax6 promotes radial glial proliferation and spindle orientation, while also facilitating neurogenesis through induction of proneural basic helix–loop–helix (bHLH) transcription factors such as Ascl1 and Neurog2 [2]. Neuronal commitment and differentiation are subsequently initiated by the bHLH transcription factors Ascl1 and Neurog2, which coordinate the transition of progenitor cells into intermediate neurogenic states [3]. Finally, neuronal maturation and survival are regulated by expression of Neurod1 and Tubb3 (class III β -tubulin), which are essential for neuronal differentiation, maturation, and synaptic function [4].

3. Figure 3: The downregulation of Neurod1, Neurog2, and Ascl1 is shown at the mRNA level; however, confirmation by immunohistochemistry and Western blot is essential for publication. Protein-level evidence would also support the claim that MO-induced inflammation impacts neurogenic fate at the translational level. Add immunohistochemistry and Western blot validation for NEUROD1, NEUROG2, and ASCL1 in embryonic brains (immunostaining or Western blot). It is also beneficial to include some of the other known components, such as Neural Stem/Progenitor Cell Maintenance, Proneural Commitment Regulators, Neuronal Differentiation and Maturation, and their modulators.

Response:

To address this concern, we assessed NEUROG2 protein expression by Western blot analysis. While NEUROG2 mRNA levels were significantly reduced in MO, NEUROG2 protein levels did not show a significant difference at E13.5.

However, we performed immunofluorescence staining of embryonic brain sections using NeuN, a marker of post-mitotic neurons. The result demonstrates a significant reduction in neuronal populations in embryos from MO compared to CON. This finding provides direct *in vivo* evidence that maternal obesity disrupts neurogenic outcomes, even in the absence of robust changes in certain proneural transcription factor protein levels.

Figure 2. **NEUROG2 protein expression in E13.5 brain.** Representative Western blot and quantification of NEUROG2 protein levels in E13.5 brain from CON and MO groups at E13.5. NEUROG2 protein levels did not differ significantly between groups. Data are presented as mean \pm SEM (n = 6).

Figure3. **MO reduces neuronal populations in the E13.5 brain in vivo.**

Representative immunofluorescence images of E13.5 mouse embryonic brain from CON and MO stained for NeuN (green), a marker of post-mitotic neurons, with DAPI (blue) labeling nuclei. Data are presented as mean \pm SEM (n = 6). **P* < 0.05. Scale bar = 400 μ m.

4. Figure 4: Please include qRT-PCR and western blots for the inflammatory genes explained.

Response:

To address this concern, we performed additional Western blot analyses to assess the expression of key inflammatory markers implicated in our study.

Specifically, we analyzed the protein levels of TNF- α and JUN, a core component of the AP-1 transcription factor complex. Both TNF- α and JUN protein expression were significantly increased in embryos from obese mothers compared to controls. These protein-level changes are consistent with our single-cell RNA sequencing and RT-qPCR results, which showed elevated expression of inflammatory cytokines and AP-1 pathway genes under maternal obesity conditions.

Figure 4. **Maternal obesity increases JUN and TNF- α protein expression in the E13.5 embryonic brain.** Representative Western blots and quantitative analysis of JUN and TNF- α protein levels in E13.5 embryonic brain tissues from CON and MO groups. Data are presented as mean \pm SEM (n = 6). * P < 0.05.

5. Fig.5: Verify the chromatin acceptability changes by Chip analysis

Response:

To validate the chromatin accessibility, we performed ChIP-qPCR. Using an antibody against c-JUN, we observed significantly greater enrichment of c-JUN binding at the *Tnf* promoter in E13.5 embryos from the MO group compared to CON. These results are consistent with our scATAC-seq findings, which revealed enhanced AP-1 motif activity and increased chromatin accessibility at the *Tnf* locus under maternal obesity conditions.

Figure 5. **Increased c-JUN binding at the *Tnf* promoter in E13.5 embryos from MO.** ChIP-qPCR showing enrichment of c-JUN binding at the *Tnf* promoter region in E13.5 embryonic tissues from CON and MO groups. Data are presented as mean \pm SEM (n = 6). * $P < 0.05$.

6. Fig. 6 verifies the results by Western blot

Response:

In Figure 6, our in vitro experiments were designed to assess the transcriptional response of neurogenic genes to inflammatory stimulation during neural differentiation. While we did not perform Western blot analyses for neurogenic transcription factors in this in vitro study, previous studies have demonstrated that TNF- α signaling suppresses neuronal differentiation, including reduced expression of neuronal markers and impaired neuronal maturation in neural progenitor cells [5-7].

Additionally, our immunofluorescence analysis of embryonic brains revealed a significant reduction in NeuN⁺ neuronal populations in embryos from obese mothers, indicating impaired neurogenesis. Together, these results support the conclusion that TNF- α -mediated inflammation negatively impacts neurogenic progression, consistent with both our in vitro transcriptional findings and established literature.

7. Another major issue is that the maternal metabolic phenotype is not sufficiently characterized. While the maternal diet is clearly defined, there is no data on maternal insulin, glucose, lipid profile including, HDL, LDL, triglycerides, total cholesterol and systemic inflammatory markers (e.g., TNF- α , IL-6, CRP), which are necessary to correlate maternal status with fetal outcomes and establish the broader physiological relevance.

Response:

To address this concern, we have expanded our analysis of the maternal metabolic phenotype. Maternal serum lipid profiles were assessed, including LDL, HDL, and triglycerides, as well as measurements of glucose, insulin, and the inflammatory marker TNF- α . We found that obese dams exhibited significantly elevated levels of LDL, HDL, triglycerides, and circulating TNF- α compared to control dams, indicating dyslipidemia and systemic inflammation under maternal obesity conditions. In contrast, maternal glucose and insulin levels did not differ significantly between groups.

Figure 6. **Maternal serum levels of LDL, HDL, triglycerides, glucose, insulin, and TNF- α .** Obese dams exhibited significantly elevated lipid levels (LDL, HDL, triglycerides) and increased circulating TNF- α compared to controls, while glucose and insulin levels were not significantly different between groups. Data are presented as mean \pm SEM (n = 6). * $P < 0.05$.

8. Additionally, the weight, age, lipid, and glucose levels of the male animals used in the experiments are also required, as variations in their weights and age have been shown to influence the metabolic status of the offspring.

Response:

In this study, male mice used for breeding were maintained on a standard control diet and were not subjected to any experimental dietary manipulation. Male breeders were age-matched and housed under identical conditions, and were randomly assigned for mating to minimize potential confounding effects related to paternal age, weight, or metabolic status. Because the experimental design specifically focused on maternal obesity as the variable of interest, paternal metabolic parameters were controlled by maintaining all male breeders under uniform conditions; detailed metabolic profiling of male breeders was not performed, as paternal variability was minimized by the experimental design.

9. Another major limitation is that the study samples embryos only at E11.5 and E13.5, which limits the temporal resolution. Key developmental events such as early neural

induction (around E9.5) and terminal differentiation or synaptic maturation (E17.5-P0) are not captured. A more complete developmental trajectory would greatly strengthen the conclusions.

Response:

The present study focuses on early embryonic neurogenesis, and E11.5–E13.5 represents a critical developmental window during which neural progenitor proliferation, proneural commitment, and early neuronal differentiation are highly active. In addition, early neural induction remains active at the developing end of the neural tube.

Thus, this developmental period enables us to investigate how maternal obesity affects early neurogenic programming. By integrating single-cell transcriptomic and epigenomic analyses with in vivo and in vitro validation at these early stages, our study captures primary molecular and cellular disruptions in neurogenesis induced by MO.

Indeed, future studies exploring later differentiation and maturation stages (E17.5–postnatal) will be important for defining the full developmental trajectory. Nevertheless, we believe that our focused analysis of early neurogenesis provides valuable mechanistic insight into how MO initiates inflammatory and transcriptional programs that impair the early neural development.

Minor concerns:

1. The raw data should be submitted, and the link is needed to verify the data which is missing.

Response:

The data for this study are available in the Gene Expression Omnibus (GEO), under accession number GSE278631.

2. Explain the neural stem cell culture media and the differentiation media clearly. The components, along with their concentrations, will facilitate the repetition of these experiments.

Response:

Neural stem cells were cultured in DMEM/F12 (1:1) supplemented with 1×B27, 20 ng/mL epidermal growth factor (EGF), and 20 ng/mL basic fibroblast growth factor (bFGF). For neuronal differentiation, cells were cultured in DMEM/F12 (1:1) supplemented with 1×B27.

3. Overall, the study is well-conceived and presents compelling preliminary evidence that maternal obesity impairs early neurogenesis through inflammation-driven transcriptional and epigenetic reprogramming. However, before submission, the concerns raised needed to be addressed.

Response:

Thank you for the positive assessment of our study and for the constructive suggestions provided. In response to these comments, we have performed extensive additional experiments and made substantial revisions to the manuscript to address all major and minor concerns. These include in vivo histological and protein-level validation, expanded characterization of maternal inflammatory and metabolic phenotypes, and mechanistic validation of AP-1–mediated regulation using CHIP-qPCR using embryonic brain. We believe that these additions significantly strengthen the rigor and impact of the study and address the issues raised.

References:

1. Saade, M. and E. Marti, *Early spinal cord development: from neural tube formation to neurogenesis*. Nat Rev Neurosci, 2025. **26**(4): p. 195-213.
2. Paridaen, J.T. and W.B. Huttner, *Neurogenesis during development of the vertebrate central nervous system*. EMBO Rep, 2014. **15**(4): p. 351-64.
3. Roybon, L., T. Hjalt, S. Stott, F. Guillemot, J.Y. Li, and P. Brundin, *Neurogenin2 directs granule neuroblast production and amplification while NeuroD1 specifies neuronal fate during hippocampal neurogenesis*. PLoS One, 2009. **4**(3): p. e4779.
4. Tutukova, S., V. Tarabykin, and L.R. Hernandez-Miranda, *The Role of Neurod Genes in Brain Development, Function, and Disease*. Front Mol Neurosci, 2021. **14**: p. 662774.
5. Johansson, S., J. Price, and M. Modo, *Effect of inflammatory cytokines on major histocompatibility complex expression and differentiation of human neural stem/progenitor cells*. Stem Cells, 2008. **26**(9): p. 2444-54.
6. Lan, X., Q. Chen, Y. Wang, B. Jia, L. Sun, J. Zheng, and H. Peng, *TNF-alpha affects human cortical neural progenitor cell differentiation through the autocrine secretion of leukemia inhibitory factor*. PLoS One, 2012. **7**(12): p. e50783.
7. Borsini, A., P.A. Zunszain, S. Thuret, and C.M. Pariante, *The role of inflammatory cytokines as key modulators of neurogenesis*. Trends Neurosci, 2015. **38**(3): p. 145-57.

Re: JP-RP-2026-289326R1 **"Maternal Obesity Induces AP-1-Mediated Inflammatory Response to Impair Embryonic Neurogenesis"** by Li Wei Chen, Md Nazmul Hossain, Yao Gao, Zhongyun Kou, Sharmeen Islam, xinrui li, Chaeyoung Shin, Jeanene Marie de Avila, Meijun Zhu, and Min Du

Dear Dr Du,

Thank you for submitting your manuscript to The Journal of Physiology. It has been assessed by a Reviewing Editor and by 2 expert referees and we are pleased to tell you that it is potentially acceptable for publication following satisfactory major revision.

Please address all the points raised and incorporate all requested revisions or explain in your Response to Referees why a change has not been made. We hope you will find the comments helpful and that you will be able to return your revised manuscript within 2 months. If your article is NOT for a Special Issue, you may have 9 months to revise. If you require an extension, please contact journal staff: jp@physoc.org. Please note that this letter does not constitute a guarantee for acceptance of your revised manuscript.

REVISION CHECKLIST:

We look forward to receiving your revised submission.

Yours sincerely,

Laura Bennet
Senior Editor
The Journal of Physiology

REQUIRED ITEMS

- Papers must comply with the Statistics Policy: https://jp.msubmit.net/cgi-bin/main.plex?form_type=display_requirements#statistics.

In summary:

- If $n \leq 30$, all data points must be plotted in the figure in a way that reveals their range and distribution. A bar graph with data points overlaid, a box and whisker plot or a violin plot (preferably with data points included) are acceptable formats.
- If $n > 30$, then the entire raw dataset must be made available either as supporting information, or hosted on a not-for-profit repository, e.g. FigShare, with access details provided in the manuscript.
- 'n' clearly defined (e.g. x cells from y slices in z animals) in the Methods. Authors should be mindful of pseudoreplication.
- All relevant 'n' values must be clearly stated in the main text, figures and tables.
- The most appropriate summary statistic (e.g. mean or median and standard deviation) must be used. Standard Error of the Mean (SEM) alone is not permitted.
- Exact p values must be stated. Authors must not use 'greater than' or 'less than'. Exact p values must be stated to three significant figures even when 'no statistical significance' is claimed.

EDITOR COMMENTS

Reviewing Editor:

Comments for Authors to ensure the paper complies with the Statistics Policy:
Some of the figures/data do not conform to the statistics policy for J Phys, including presenting all data and exact P values. Please carefully check the statistics policy above.

Comments to the Author:

Thank you for your revised manuscript. Reviewer 1 has highlighted a number of points that still require addressing. Please respond to these, including any additional experiments suggested. This includes, but is not limited to, a region-specific validation demonstrating local changes within the developing brain, such as detailed immunohistochemical analyses and/or protein-level assays from brain tissue. Please also ensure that any new data, and all changes indicated within the cover letter, are fully integrated into the revised manuscript. Please also note that some of the figures/data do not conform to the statistics policy for J Phys, including presenting all data and exact P Values.

Senior Editor:

Comments for Authors to ensure the paper complies with the Statistics Policy:

See RE comments, figures are not presenting all data points when they could and $P <$ rather than $p =$ in places.

Comments to the Author:

Thank you for your resubmission. It is important that you fully address the requests made by reviewer one to assure us about the outcomes and impact of the data.

REFeree COMMENTS

Referee #1:

The authors have addressed the points raised in the previous review; however, several major concerns remain.

1) The manuscript states that scRNA-seq was performed using "embryos," but it is unclear whether the analysis was conducted on whole embryos or on specific tissues. As previously pointed out, it is essential to clearly indicate that the scRNA-seq data were derived from whole embryos, if this is the case. This point should be explicitly described in the Methods and also clarified in the main text, as it is critical for correct interpretation of the results.

2) Regarding the scRNA-seq data in Figure 1, particularly Fig. 1E, there appear to be substantial differences in overall cell population composition between the Control and MO groups. For example, fibroblast populations seem to be markedly increased in the MO condition. It is possible that changes in fetal body weight or overall embryo size altered the relative proportions of different cell types at the whole-embryo level. Such population shifts could secondarily reduce the relative proportion of cells categorized as neurons or neural progenitors, without necessarily reflecting a true impairment of neurogenesis. In this context, the statement that the scRNA-seq results are "indicating a profound impairment in neurogenesis" appears to be an overinterpretation based solely on whole-embryo scRNA-seq data.

3) In the previous round of review, I requested in vivo validation of the effects on the nervous system. In response, immunohistochemical data were presented in the rebuttal letter; however, critical issues remain. It is unclear which region of the embryonic brain is being analyzed. If the images are intended to represent the developing cortex at this stage, the reported NeuN distribution is difficult to reconcile with known developmental anatomy, as a clear laminar organization would be expected. The brain region examined is not specified, and the anatomical context is insufficiently documented.

In addition, the relationship between the fetal brain structure and the scale bar appears unnatural, making it difficult to assess spatial context. Clear anatomical landmarks and low-magnification overview images should be provided.

4) Furthermore, although additional experiments such as Western blotting are mentioned, these data are not included in the revised manuscript. Given that these experiments are critical, they should be incorporated into the revised version.

5) For Figures 3 and 6, individual data points should be shown rather than only summary statistics. Displaying individual values is essential for evaluating data variability and robustness.

6) The statement "At these early developmental stages, the nervous system is mainly present as neural tubes and associated neural crest" is not strictly accurate. At E11.5-E13.5, extensive and regionally specialized neurogenesis is actively occurring in the developing brain, particularly in the forebrain. This description should therefore be revised with greater anatomical precision and caution.

7) Overall, because the scRNA-seq analysis is based on whole embryos, changes in global cell population composition may confound the apparent changes observed in neural cell populations. Therefore, conclusions regarding neurodevelopmental impairment should be interpreted with caution. To strengthen the study, additional region-specific validation demonstrating local changes within the developing brain—such as detailed immunohistochemical analyses and/or protein-level assays from brain tissue—would be important. At present, these data are limited or not fully integrated into the revised version.

Referee #2:

The authors have performed most of the experiments and responded to most of the comments raised in the earlier version.

While most of the issues are now addressed, I am not able to see the figure changes in the figures which are uploaded with the revised version. They should be included in the revised figures.

I have another minor comment: the data points must be shown for all the figures. The older figures lack these.

END OF COMMENTS

Referee Report

The authors have performed most of the experiments and responded to most of the comments raised in the earlier version.

While most of the issues are now addressed, I cannot see the figure changes in the figures which are uploaded with the revised version. They should be included in the revised figures.

I have another minor comment: the data points must be shown for all the figures. The older figures lack these.

Dear Dr Du,

Thank you for submitting your manuscript to The Journal of Physiology. It has been assessed by a Reviewing Editor and by 2 expert referees and we are pleased to tell you that it is potentially acceptable for publication following satisfactory major revision.

Please address all the points raised and incorporate all requested revisions or explain in your Response to Referees why a change has not been made. We hope you will find the comments helpful and that you will be able to return your revised manuscript within 2 months. If your article is NOT for a Special Issue, you may have 9 months to revise. If you require an extension, please contact journal staff: jp@physoc.org. Please note that this letter does not constitute a guarantee for acceptance of your revised manuscript.

REVISION CHECKLIST:

We look forward to receiving your revised submission.

Yours sincerely,

Laura Bennet
Senior Editor
The Journal of Physiology

REQUIRED ITEMS

- Papers must comply with the Statistics Policy: https://jp.msubmit.net/cgi-bin/main.plex?form_type=display_requirements#statistics.

In summary:

- If n less than or equal to 30, all data points must be plotted in the figure in a way that reveals their range and distribution. A bar graph with data points overlaid, a box and whisker plot or a violin plot (preferably with data points included) are acceptable formats.

- If $n > 30$, then the entire raw dataset must be made available either as supporting information, or hosted on a not-for-profit repository, e.g. FigShare, with access details provided in the manuscript.

- 'n' clearly defined (e.g. x cells from y slices in z animals) in the Methods. Authors should be mindful of pseudoreplication.

- All relevant 'n' values must be clearly stated in the main text, figures and tables.

- The most appropriate summary statistic (e.g. mean or median and standard deviation) must be used. Standard Error of the Mean (SEM) alone is not permitted.

- Exact p values must be stated. Authors must not use 'greater than' or 'less than'. Exact p values must be stated to three significant figures even when 'no statistical significance' is claimed.

EDITOR COMMENTS

Reviewing Editor:

Comments for Authors to ensure the paper complies with the Statistics Policy:
Some of the figures/data do not conform to the statistics policy for J Phys, including presenting all data and exact P values. Please carefully check the statistics policy above.

Comments to the Author:

Thank you for your revised manuscript. Reviewer 1 has highlighted a number of points that still require addressing. Please respond to these, including any additional experiments suggested. This includes, but is not limited to, a region-specific validation demonstrating local changes within the developing brain, such as detailed immunohistochemical analyses and/or protein-level assays from brain tissue. Please also ensure that any new data, and all changes indicated within the cover letter, are fully integrated into the revised manuscript. Please also note that some of the figures/data do not conform to the statistics policy for J Phys, including presenting all data and exact P Values.

Response:

We thank the Reviewing Editor for the careful evaluation of our revised manuscript and for the guidance. We have now integrated revised and newly generated figures into the manuscript, revised the figures to present individual data points in accordance with the J Phys statistics policy, and addressed all comments raised by the reviewers. We believe that these revisions strengthen the impact of the study and fully address the reviewers' concerns.

Senior Editor:

Comments for Authors to ensure the paper complies with the Statistics Policy:
See RE comments, figures are not presenting all data points when they could and $P <$ rather than $p =$ in places.

Comments to the Author:

Thank you for your resubmission. It is important that you fully address the requests made by reviewer one to assure us about the outcomes and impact of the data.

Response:

We thank the Senior Editor for the careful evaluation of our revised manuscript and for the guidance regarding the statistics policy. We have revised all relevant figures to display individual data points and updated the statistical reporting throughout the manuscript to present exact P values in compliance with the J Phys guidelines.

In addition, we have now fully addressed all requests raised by the reviewers, and we believe that these revisions strengthen the impact of the study.

REFEREE COMMENTS

Referee #1:

The authors have addressed the points raised in the previous review; however, several major concerns remain.

1) The manuscript states that scRNA-seq was performed using "embryos," but it is unclear whether the analysis was conducted on whole embryos or on specific tissues. As previously pointed out, it is essential to clearly indicate that the scRNA-seq data were derived from whole embryos, if this is the case. This point should be explicitly described in the Methods and also clarified in the main text, as it is critical for correct interpretation of the results.

Response:

Thank you for this comment. We have now clarified in both the Methods and the main text that single-cell RNA sequencing was performed on whole embryos as previously described (Hossain et al., 2024). Neural progenitor and neuronal populations were clustered computationally based on their transcriptomes. .

2) Regarding the scRNA-seq data in Figure 1, particularly Fig. 1E, there appear to be substantial differences in overall cell population composition between the Control and MO groups. For example, fibroblast populations seem to be markedly increased in the MO condition. It is possible that changes in fetal body weight or overall embryo size altered the relative proportions of different cell types at the whole-embryo level. Such population shifts could secondarily reduce the relative proportion of cells categorized as neurons or neural progenitors, without necessarily reflecting a true impairment of neurogenesis. In this context, the statement that the scRNA-seq results are "indicating a profound impairment in neurogenesis" appears to be an overinterpretation based solely on whole-embryo scRNA-seq data.

Response:

Because cells were first clustered based on their individual transcriptomes and only virtually-sorted progenitors and neurogenic cells were used, most analyses were not affected by the presence of fibroblasts. To address the reviewer's concern, we have revised the text to avoid overinterpretation and now describe these findings as alterations in neurogenic cell populations under MO.

3) In the previous round of review, I requested in vivo validation of the effects on the nervous system. In response, immunohistochemical data were presented in the rebuttal letter; however, critical issues remain. It is unclear which region of the embryonic brain is being analyzed. If the images are intended to represent the developing cortex at this stage, the reported NeuN distribution is difficult to reconcile with known developmental anatomy, as a clear laminar organization would be expected. The brain region examined is not specified, and the anatomical context is insufficiently documented.

In addition, the relationship between the fetal brain structure and the scale bar appears unnatural, making it difficult to assess spatial context. Clear anatomical landmarks and low-magnification overview images should be provided.

Response:

We have now clarified the brain region analyzed and revised both the figure and legend accordingly. Specifically, the immunofluorescence images correspond to coronal sections of the ganglionic eminence in the ventral telencephalon at E13.5.

We have added low-magnification overview images with boxed regions indicating the areas selected for quantification, and verified that the scale bars are correct.

4) Furthermore, although additional experiments such as Western blotting are mentioned, these data are not included in the revised manuscript. Given that these experiments are critical, they should be incorporated into the revised version. 5) For Figures 3 and 6, individual data points should be shown rather than only summary statistics. Displaying individual values is essential for evaluating data variability and robustness.

Response:

The revised figures have now been included in the updated version.

Individual data points have been added to all relevant figures.

6) The statement "At these early developmental stages, the nervous system is mainly present as neural tubes and associated neural crest" is not strictly accurate. At E11.5-E13.5, extensive and regionally specialized neurogenesis is actively occurring in the developing brain, particularly in the forebrain. This description should therefore be revised with greater anatomical precision and caution.

Response:

We have removed the statement “At these early developmental stages, the nervous system is mainly present as neural tubes and associated neural crest”.

7) Overall, because the scRNA-seq analysis is based on whole embryos, changes in global cell population composition may confound the apparent changes observed in neural cell populations. Therefore, conclusions regarding neurodevelopmental impairment should be interpreted with caution. To strengthen the study, additional region-specific validation demonstrating local changes within the developing brain-such as detailed immunohistochemical analyses and/or protein-level assays from brain tissue-would be important. At present, these data are limited or not fully integrated into the revised version.

Response:

To directly evaluate local neurogenic changes within the developing brain, we have now fully integrated region-specific in vivo and protein-level validation using embryonic brain tissue.

Specifically, we performed immunofluorescence analysis of the ganglionic eminence at E13.5 demonstrating a significant reduction in NeuN⁺ neuronal populations in the MO, showing that MO impairs neurogenesis within the developing brain. We have also revised the manuscript to avoid overinterpretation of scRNA-seq results.

Referee #2:

The authors have performed most of the experiments and responded to most of the comments raised in the earlier version.

While most of the issues are now addressed, I am not able to see the figure changes in the figures which are uploaded with the revised version. They should be included in the revised figures.

I have another minor comment: the data points must be shown for all the figures. The older figures lack these.

Response:

Thank you for your comments. The revised figures have now been included in the updated version. In addition, individual data points have been added to all relevant figures.

Reference

Hossain, M. N., Gao, Y., Li, X., Zhao, L., Liu, X., Marie de Avila, J., Zhu, M. J., & Du, M. (2024). Single-cell RNA transcriptomics in mice reveals embryonic origin of fibrosis due to maternal obesity. *EBioMedicine*, 109, 105421. <https://doi.org/10.1016/j.ebiom.2024.105421>

Dear Dr Du,

Re: JP-RP-2026-289326R2 "**Maternal Obesity Induces AP-1-Mediated Inflammatory Response to Impair Embryonic Neurogenesis**" by Li Wei Chen, Md Nazmul Hossain, Yao Gao, Zhongyun Kou, Sharmeen Islam, xinrui li, Chaeyoung Shin, Jeanene Marie de Avila, Meijun Zhu, and Min Du

We are pleased to tell you that your paper has been accepted for publication in The Journal of Physiology.

Please see below for a minor amendment that should be made at proof stage.

Yours sincerely,

Laura Bennet
Senior Editor
The Journal of Physiology

IMPORTANT POINTS TO NOTE FOLLOWING ACCEPTANCE OF YOUR PAPER:

- **IMPORTANT NOTICE ABOUT OPEN ACCESS:** To assist authors whose funding agencies mandate immediate public access to published research findings, The Journal of Physiology allows authors to pay an Open Access (OA) fee to have their papers made freely available immediately on publication.

The Corresponding Author will receive an email from Wiley with details on how to register or log in to Wiley Authors where you will be able to place an order.

- You can check if your funder or institution has a Wiley Open Access Account here:
<https://authors.wiley.com/author-resources/Journal-Authors/open-access/author-compliance-tool.html>

- You can help your research get the attention it deserves! Check out Wiley's free Promotion Guide for best-practice recommendations for promoting your work at: www.wileyauthors.com/eo/guide. You can learn more about Wiley Editing Services which offers professional video, design, and writing services to create shareable video abstracts, infographics, conference posters, lay summaries, and research news stories for your research at: www.wileyauthors.com/eo/promotion.

- If you would like to receive our 'Research Roundup', a monthly newsletter highlighting the cutting-edge research published in The Physiological Society's family of journals (The Journal of Physiology, Experimental Physiology, Physiological Reports, The Journal of Nutritional Physiology and The Journal of Precision Medicine: Health and Disease), please click this link, fill in your name and email address and select 'Research Roundup':
<https://www.physoc.org/journals-and-media/membernews>

EDITOR COMMENTS

Reviewing Editor:

Thank you for your submission. I note one small text error.

'Maternal blood was collected at E13.5 by cardiopuncture under carbon dioxide anesthetized.'

Please change to

'Maternal blood was collected at E13.5 by cardiopuncture under carbon dioxide anesthesia. '

Thank you

REFeree COMMENTS

Referee #1:

The authors addressed the points raised in my previous review. The revisions have strengthened the manuscript, and I have no additional comments.

Referee #2:

No further comments.